# Non-linear machine learning models incorporating SNPs and PRS improve polygenic prediction in diverse human populations

Michael Elgart [1,2,19✉], Genevieve Lyons [1,3,19], Santiago Romero-Brufau [3,4], Nuzulul Kurniansyah[1], Jennifer A. Brody [5], Xiuqing Guo [6], Henry J. Lin [6], Laura Raffield [7], Yan Gao[8], Han Chen[9,10], Paul de Vries[9], Donald M. Lloyd-Jones[11], Leslie A. Lange[12], Gina M. Peloso [13], Myriam Fornage [9,14], Jerome I. Rotter[6], Stephen S. Rich [15], Alanna C. Morrison [9], Bruce M. Psaty [16], Daniel Levy[17,18], Susan Redline [1,2], the NHLBI's Trans-Omics in Precision Medicine (TOPMed) Consortium* & Tamar Sofer [1,2,3✉]

Polygenic risk scores (PRS) are commonly used to quantify the inherited susceptibility for a trait, yet they fail to account for non-linear and interaction effects between single nucleotide polymorphisms (SNPs). We address this via a machine learning approach, validated in nine complex phenotypes in a multi-ancestry population. We use an ensemble method of SNP selection followed by gradient boosted trees (XGBoost) to allow for non-linearities and inter-action effects. We compare our results to the standard, linear PRS model developed using PRSice, LDpred2, and lassosum2. Combining a PRS as a feature in an XGBoost model results in a relative increase in the percentage variance explained compared to the standard linear PRS model by 22% for height, 27% for HDL cholesterol, 43% for body mass index, 50% for sleep duration, 58% for systolic blood pressure, 64% for total cholesterol, 66% for triglycerides, 77% for LDL cholesterol, and 100% for diastolic blood pressure. Multi-ancestry trained models perform similarly to specific racial/ethnic group trained models and are consistently superior to the standard linear PRS models. This work demonstrates an effective method to account for non-linearities and interaction effects in genetics-based prediction models.

[1] Division of Sleep and Circadian Disorders, Brigham and Women's Hospital, Boston, MA, USA. [2] Department of Medicine, Harvard Medical School, Boston, MA, USA. [3] Department of Biostatistics, Harvard T.H. Chan School of Public Health, Boston, MA, USA. [4] Department of Medicine, Mayo Clinic, Rochester, MN, USA. [5] Cardiovascular Health Research Unit, Department of Medicine, University of Washington, Seattle, WA, USA. [6] The Institute for Translational Genomics and Population Sciences, Department of Pediatrics, The Lundquist Institute for Biomedical Innovation at Harbor-UCLA Medical Center, Torrance, CA, USA. [7] Department of Genetics, University of North Carolina, Chapel Hill, NC, USA. [8] The Jackson Heart Study, University of Mississippi Medical Center, Jackson, MS, USA. [9] Human Genetics Center, Department of Epidemiology, Human Genetics, and Environmental Sciences, School of Public Health, The University of Texas Health Science Center at Houston, Houston, TX, USA. [10] Center for Precision Health, School of Biomedical Informatics, The University of Texas Health Science Center at Houston, Houston, TX, USA. [11] Department of Preventive Medicine, Northwestern University, Chicago, IL, USA. [12] Department of Medicine, University of Colorado Denver, Anschutz Medical Campus, Aurora, CO, USA. [13] Department of Biostatistics, Boston University School of Public Health, Boston, MA, USA. [14] Brown Foundation Institute of Molecular Medicine, McGovern Medical School, University of Texas Health Science Center at Houston, Houston, TX, USA. [15] Center for Public Health Genomics, University of Virginia School of Medicine, Charlottesville, VA, USA. [16] Cardiovascular Health Research Unit, Departments of Medicine, Epidemiology, and Health Services, University of Washington, Seattle, WA, USA. [17] The Population Sciences Branch of the National Heart, Lung and Blood Institute, Bethesda, MD, USA. [18] The Framingham Heart Study, Framingham, MA, USA. [19]These authors contributed equally: Michael Elgart, Genevieve Lyons. *A list of authors and their affiliations appears at the end of the paper. ✉email: melgart@bwh.harvard.edu; tsofer@bwh.harvard.edu

In the last few years, genetics-based trait prediction using polygenic risk scores (PRS) have become increasingly popular. PRS are calculated as weighted sums of allele counts for variants that are associated with an outcome of interest and are used to quantify the inherited susceptibility for a given trait or disease[1]. Traditionally, genome-wide association studies (GWAS) are used to identify the univariate relationships between single nucleotide polymorphisms (SNPs) and a given phenotype. These univariate relationships are then used to construct the PRS[2].

The prediction models that use PRS are generally able to explain only a small percentage of the observed variance for a given trait[2], which could be due to several factors. Because they rely on univariate effect sizes derived from linear GWAS models, standard PRS as defined above do not account for potential nonlinearities in the association between the genetic data and the outcome of interest. Additionally, additive PRS models do not account for interactions between SNPs, which are known to exist[3]. One common strategy employed during the SNP selection stage of PRS construction is clumping, to exclude SNPs within a predefined distance of one another and levels of linkage disequilibrium (LD). Potential interactions are not usually taken into account by this approach—as in haplotypes[4] or epistatic effects[5] both inside and outside the clumping region. Examples of strong haplotypes effects that may not be captured by a clump-and threshold approach are *APOE* (associated with Alzheimer's disease)[6] and *APOL1* (associated with chronic kidney disease)[7] haplotypes. Many other haplotypes with lower effect sizes may be yet unknown and harder to detect. In addition, clumping may not select causal variants or optimally tag SNPs for the population at hand. Other modern methods such as LDPred2[8] and PRS-CS[9] construct PRS while using GWAS results in combination with an LD reference panel to estimate joint SNP effects, typically using many SNPs in LD with each other from a given association region, potentially overcoming the above limitations of the clumping-based approaches. However, such methods still may not capture haplotype effects because haplotype inference requires phased genetic data. Another limitation of PRSs is that effect sizes based on summary statistics from a GWAS conducted in one population may not be optimal for a different population. Specifically, PRS performance is known to be affected by the population in which the GWAS was conducted, and PRS may not generalize well to different populations[10–12].

Some of the challenges of PRS modeling can be addressed using advanced machine learning (ML) methodologies. Many ML algorithms such as random forests, gradient boosted trees, and neural networks are explicitly nonlinear, and allow interaction between features. Gradient boosted trees, for example, allow for the effect size of a given SNP to vary depending on the presence of an allele of a different SNP[13]. Accordingly, ML methods have been used successfully to improve the prediction of complex phenotypes using genetic data[14]. For example, a study employing random forests to predict type 2 diabetes found that it outperformed linear models, such as support vector machines[15]. Gradient boosted trees have been used to predict breast cancer risk by first identifying nonlinear SNP-SNP interactions using XGBoost or networks and then using support vector machines for discrimination, which resulted in increased mean average precision when compared to generalized linear models[16,17]. Other studies have found that deep neural networks outperform linear models for a wide range of diseases, but not all[18]. Finally, while ML models do not explicitly allow for generalization to non-sampled populations, we hypothesize that a large and ancestry diverse cohort would improve genetic prediction across populations.

Here, we explore the use of genetic data in prediction of nine complex phenotypes: six established cardiovascular disease risk factors (total cholesterol levels[19], LDL cholesterol, HDL cholesterol, triglycerides[20], systolic blood pressure[21], and diastolic blood pressure), sleep duration, a phenotype of lower heritability that is also associated with cardiovascular disease[22,23], body mass index, and height, a highly heritable and well-studied phenotype. For each of these nine complex phenotypes, we develop ensemble machine learning models for genetic trait prediction accounting for interactions, trained on a multi-ethnic dataset from the National Heart Lung and Blood Institute's Trans-Omics in Precision Medicine (TOPMed) consortium[24]. We examine the accuracy of the results to linear models with clump-and threshold (C + T) PRS (using PRSice), LDpred2 PRS, and lassosum2 PRS, and then explicitly compare ML models that allow for interactions and nonlinear effects to those that do not. Finally, we assess the accuracy of the predictions for the ML models and the linear PRS models among White, Black, and Hispanic/Latino race/ethnic groups.

## Results

We used a multi-ethnic dataset from TOPMed containing 29,063 genotyped individuals from eight distinct cohorts (JHS, FHS, HCHS/SOL, ARIC, CHS, MESA, CFS, and CARDIA) to train nonlinear polygenic risk prediction models in diverse populations for nine complex phenotypes: triglycerides, total cholesterol, LDL cholesterol, HDL cholesterol, systolic blood pressure, diastolic blood pressure, sleep duration, body mass index, and height. Table 1 characterizes these phenotypes and covariates across the pooled and race/ethnicity-stratified training dataset based on unrelated individuals. We evaluate the models on an independent test dataset of 5009 individuals (Supplementary Table 1). Results based on a training dataset that included related individuals (and still being unrelated to the test dataset) were similar. Summary statistics for this dataset are provided in Supplementary Table 2 and results are provided in Supplementary Table 3.

**PRSice, LDpred2, and Lassosum2 linear PRS results**. We compared the percentage variance explained (PVE) of the best-performing PRSice-based PRS to the best-performing LDpred2 and lassosum2 PRS in linear PRS models. Measured in relative PVE increase, LDpred2 performed better than PRSice for four of the nine phenotypes: height (13% higher), sleep duration (25% higher), systolic blood pressure (30% higher), and diastolic blood pressure (183% higher). Lassosum2 was superior to LDpred2 and PRSice for two phenotypes: triglycerides (12% higher than PRSice PRS) and body mass index (15% higher than PRSice PRS). PRSice performed better than LDpred2 and lassosum2 for the remaining three phenotypes: HDL cholesterol (33% higher than LDpred2), LDL cholesterol (67% higher than LDpred2), and total cholesterol (121% higher than LDpred2) (Fig. 1).

For the PRS calculated with PRSice, the results were relatively invariant to the changes in the hyperparameters, rarely differing by more than 1%. The best-performing model for seven out of nine of the phenotypes had a clumping $R^2$ value of 0.1 and a clumping window of 500 kb. For the PRS calculated with LDpred2, however, the model was very sensitive to both the SNP selection and the method of LDpred2-inf compared to LDpred2-auto and LDpred2-grid. LDpred2-auto was superior to LDpred2-inf and LDpred2-grid for seven out of nine phenotypes; and using the top 1 million SNPs was superior to using the clumped SNPs for all nine phenotypes (Supplementary Fig. 1).

We selected the best-performing PRS (PRSice, LDpred2, or lassosum2) in the training dataset, and we used this PRS for purposes of comparison to XGBoost as well as included it in the XGBoost as a covariate in another model.

**Table 1 Summary statistics of phenotypes used in the training dataset.**

| Characteristic | Black (N = 7601) | Hispanic/Latino (N = 7320) | White (N = 14142) | Overall (N = 29063) |
|---|---|---|---|---|
| Sex | | | | |
| Male | 3066 (40.3%) | 3088 (42.2%) | 6432 (45.5%) | 12586 (43.3%) |
| Female | 4535 (59.7%) | 4232 (57.8%) | 7710 (54.5%) | 16477 (56.7%) |
| Age | | | | |
| Mean (SD) | 50.6 (16.9) | 48.2 (14.3) | 50.2 (16.4) | 49.8 (16.1) |
| Median [Min, Max] | 52.0 [2.00, 93.0] | 49.0 [5.00, 86.0] | 51.0 [3.00, 98.0] | 51.0 [2.00, 98.0] |
| Triglycerides | | | | |
| Mean (SD) | 106 (69.1) | 135 (96.0) | 125 (82.2) | 124 (84.2) |
| Median [Min, Max] | 90.0 [16.0, 1930] | 113 [20.0, 1670] | 106 [17.0, 1600] | 103 [16.0, 1930] |
| Missing | 2598 (34.2%) | 1316 (18.0%) | 3073 (21.7%) | 6987 (24.0%) |
| Total cholesterol | | | | |
| Mean (SD) | 198 (41.8) | 200 (43.2) | 205 (39.2) | 202 (41.0) |
| Median [Min, Max] | 196 [74.0, 450] | 197 [62.0, 526] | 202 [77.8, 594] | 199 [62.0, 594] |
| Missing | 2598 (34.2%) | 1316 (18.0%) | 3073 (21.7%) | 6987 (24.0%) |
| Systolic blood pressure | | | | |
| Mean (SD) | 127 (20.9) | 121 (17.2) | 118 (17.1) | 121 (18.5) |
| Median [Min, Max] | 123 [73.0, 246] | 119 [77.0, 218] | 116 [67.0, 227] | 118 [67.0, 246] |
| Missing | 1944 (25.6%) | 1589 (21.7%) | 2972 (21.0%) | 6505 (22.4%) |
| Sleep duration | | | | |
| Mean (SD) | 6.50 (1.51) | 7.73 (1.52) | 7.09 (1.16) | 7.15 (1.44) |
| Median [Min, Max] | 6.00 [1.00, 16.5] | 7.79 [2.00, 13.4] | 7.00 [1.00, 16.0] | 7.00 [1.00, 16.5] |
| Missing | 2352 (30.9%) | 411 (5.6%) | 4468 (31.6%) | 7231 (24.9%) |
| Height | | | | |
| Mean (SD) | 168 (10.4) | 163 (9.24) | 168 (10.3) | 167 (10.3) |
| Median [Min, Max] | 168 [85.7, 207] | 162 [116, 194] | 168 [94.0, 203] | 166 [85.7, 207] |
| Diastolic blood pressure | | | | |
| Mean (SD) | 109 (44.2) | 90.5 (36.7) | 88.4 (36.2) | 94.3 (39.5) |
| Median [Min, Max] | 85.5 [18.0, 267] | 76.0 [40.0, 256] | 74.7 [18.0, 246] | 77.0 [18.0, 267] |
| Missing | 236 (3.1%) | 9 (0.1%) | 308 (2.2%) | 553 (1.9%) |
| HDL cholesterol | | | | |
| Mean (SD) | 52.4 (14.9) | 49.1 (13.3) | 52.1 (16.0) | 51.4 (15.1) |
| Median [Min, Max] | 50.0 [15.4, 162] | 47.0 [13.0, 141] | 50.0 [9.63, 143] | 49.0 [9.63, 162] |
| Missing | 328 (4.3%) | 7 (0.1%) | 710 (5.0%) | 1045 (3.6%) |
| LDL cholesterol | | | | |
| Mean (SD) | 123 (38.1) | 122 (36.7) | 125 (36.1) | 124 (36.8) |
| Median [Min, Max] | 120 [11.6, 435] | 120 [23.8, 417] | 123 [13.8, 505] | 121 [11.6, 505] |
| Missing | 376 (4.9%) | 143 (2.0%) | 877 (6.2%) | 1396 (4.8%) |
| BMI | | | | |
| Mean (SD) | 30.0 (7.19) | 30.1 (6.29) | 26.3 (4.99) | 28.2 (6.25) |
| Median [Min, Max] | 28.9 [12.7, 91.8] | 29.1 [14.9, 70.3] | 25.6 [11.6, 66.6] | 27.2 [11.6, 91.8] |
| Missing | 6 (0.1%) | 9 (0.1%) | 8 (0.1%) | 23 (0.1%) |

Mean, Median, and percent of missing data for the phenotypes and covariates (sex and age) used in this study. Most missing values for systolic blood pressure, total cholesterol, and triglycerides are due to medication use. All the phenotypes are presented for the whole database as well as stratified by race/ethnicity (Black, White, and Hispanic/Latino). Summary statistics for the test dataset are provided in Supplementary Table 3.

**XGBoost outperforms linear models for the prediction of complex phenotypes.** We constructed four models of increasing complexity for each of the nine phenotypes (Fig. 2). Each phenotype was adjusted for the covariates and the residuals were rank-normalized. Each model was then fine-tuned to predict the residuals from genetic SNP data. The four models employed in this study were a linear model using the best-performing PRS (PRSice, LDpred2, or lassosum2), LASSO, XGBoost, and XGBoost with PRS. The number of SNPs selected for each algorithm for each phenotype listed in Table 2. The hyperparameters selected for each model through cross-validation are listed in Supplementary Table 4.

Figure 3 depicts the PVE across different prediction models and phenotypes. The linear PRS model usually outperformed the LASSO model, except for total cholesterol and LDL cholesterol, even though the LASSO model used only 19% and 12% of the SNPs used by the PRS for total cholesterol and LDL cholesterol, respectively. The XGBoost algorithm trained directly on SNPs (XGBoost alone), outperforms linear PRS models for almost all

phenotypes. The notable exceptions are body mass index and height; for these phenotypes, it substantially underperformed from the best-performing PRS model.

Supplementary Table 5 provides results from the secondary analysis comparing linear PRS model using SNPs with $p$ value $<10^{-4}$ to the XGBoost alone model. This comparison more directly assesses the effects of non-linearities and of potential overfitting of the XGBoost model, as the two compared models contain the same set of candidate SNPs. The results were qualitatively similar to those in the primary analysis: in most cases, the XGBoost alone model outperformed the linear PRS model, but not for BMI, height, and HDL cholesterol. The latter two had a larger number of SNPs, likely leading to overfitting (Table 2).

We performed two experiments to test the benefit of using LASSO to select SNPs prior to the XGBoost model. In Supplementary Table 6, we report results from random SNP selection as a baseline for four phenotypes (total cholesterol, triglycerides, LDL cholesterol, and HDL cholesterol). LASSO-selected SNPs are superior to random selection in the XGBoost

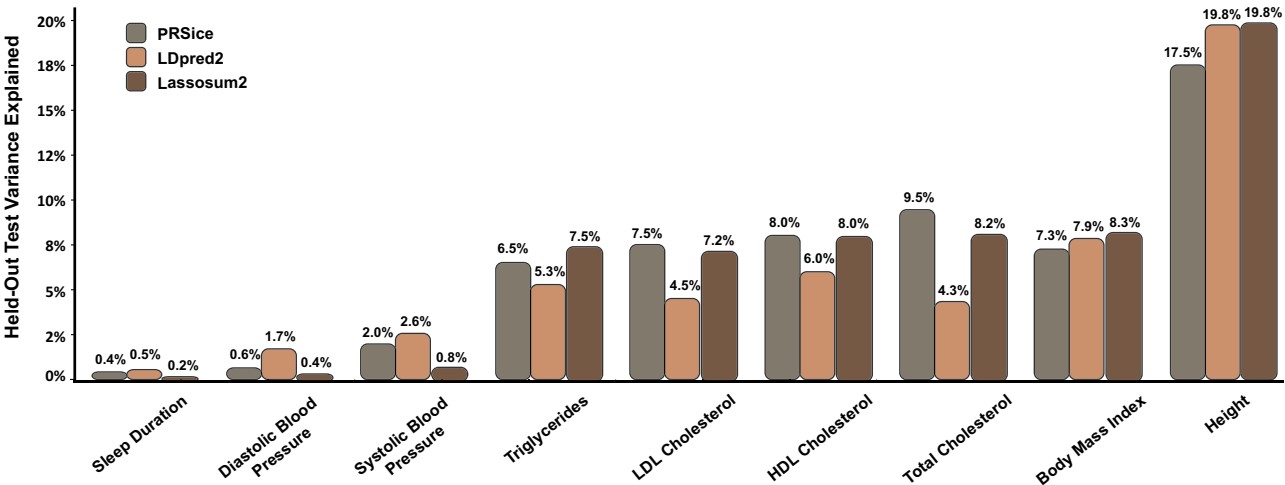

**Fig. 1 PRSice, LDpred2, and Lassosum2 Linear PRS results.** Best-performing PRSice (gray) compared to best-performing LDpred2 (orange) and best Lassosum2 (brown) across the hyperparameters tuned using the training data.

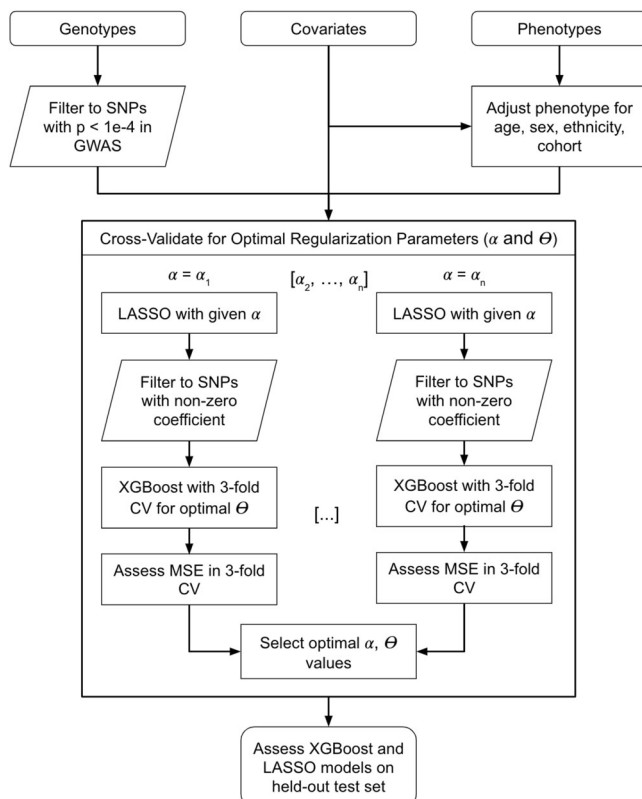

**Fig. 2 Flow chart of ensemble model structure.** The model relies on jointly training the LASSO and XGBoost model to identify the optimal value for the L1 regularization parameter and the number of boosting steps. CV indicates cross-validation, $\alpha$ refers to the regularization parameter, and $\Theta$ is the number of boosted trees for XGBoost. The optimal values for these hyperparameters were selected using threefold CV for the mean squared error of the XGBoost model.

Alone model, with relative PVE increase of 20–175%. For XGBoost with PRS, the increase from LASSO is more attenuated, at only 7–21% higher than random selection. In Supplementary Table 7, we report results from using SNPs selected into lassosum2 PRS for LDL cholesterol. The test EVR for the XGBoost model with the lassosum2-selected SNPs was 9.0% compared to 13.3% when using the LASSO SNP selection model.

We performed three additional sensitivity analyses to test the stability of our results. First, we trained our models on clumped SNPs rather than all SNPs. Supplementary Table 8 shows that clumping prior to running the LASSO and XGBoost model does not meaningfully change our results. Second, we tested the use of the genetic PCs as covariates by removing them from the training data prior to training the models. Supplementary Table 9 shows that the test set PVEs are slightly lower than those of the PRS model that does include the genetic PCs as covariates. Finally, we performed cross-validation for the optimal regularization term with respect to the LASSO loss function, rather than the joint training scheme with XGBoost. Supplementary Table 10 shows that, for the LASSO model, the results are slightly improved; however, they are not directly comparable to the XGBoost models as they include different variants.

**Modeling of non-linearities and interactions among SNPs improves the prediction of complex human phenotypes**. The nonlinear XGBoost algorithm outperforms the linear LASSO when trained on the same SNP set (Fig. 3 gray vs teal) for all phenotypes. The improved performance may stem either from modeling nonlinear genetic effects or interactions between SNPs, or both, since both of these are addressed by the algorithm[13]. However, both XGBoost and the LASSO sometimes underperformed relative to the linear PRS models based on genome-wide SNPs, likely because the PRS was able to combine information from more SNPs. Notably, for height, the XGBoost alone model had only 8.8 PVE while the PRS had 19.8 PVE. This could also be due to overfitting to the training dataset. To combine the advantages of a genome-wide PRS and of the XGBoost accounting for non-linearities and interactions, we constructed an additional, XGBoost with PRS, model that included both the individual SNPs as shown in Fig. 2, and also the best-performing PRS (PRSice, LDpred2, or lassosum2). Indeed, we see that this model substantially outperforms the linear PRS and LASSO models, as well as XGBoost alone, for all phenotypes, providing a strong indication for nonlinear effects and/or genotype by genotype interactions. Substantial improvements are observed for all phenotypes. Specifically, compared to the linear PRS baseline, the XGBoost with PRS showed a relative improvement to the PVE by 22% for height, 27% for HDL cholesterol, 43% for body mass index, 50% for sleep duration, 58% for systolic blood pressure, 64% for total cholesterol, 66% for triglycerides, 77% for LDL cholesterol, and 100% for diastolic blood pressure.

**Table 2 Number of SNPs selected through cross-validation for the PRS and XGBoost Model.**

| Phenotype | XGBoost alone | LASSO | XGBoost with PRS | PRS | Lassosum2 |
|---|---|---|---|---|---|
| Sleep duration | 35 | 35 | 36 | 1M | 140,507 |
| Diastolic blood pressure | 297 | 297 | 298 | 1M | 197,039 |
| Systolic blood pressure | 38 | 38 | 27 | 1M | 249,700 |
| Triglycerides | 186 | 186 | 109 | 6799 | 8035 |
| LDL cholesterol | 727 | 727 | 429 | 5825 | 2056 |
| HDL cholesterol | 6746 | 6746 | 168 | 7694 | 10,651 |
| Total cholesterol | 1181 | 1181 | 84 | 6258 | 4340 |
| Body mass index | 44 | 44 | 51 | 1M | 51,133 |
| Height | 4807 | 4807 | 559 | 1M | 59,714 |

Displayed are number of SNPs selected for each of the phenotypes in the four models in this study: PRS (best-performing PRS from PRSice, LDpred2, or lassosum2), XGBoost alone, LASSO (which has the same number of variants as in the XGBoost alone model, because the LASSO selected the variants used by XGBoost), and XGBoost with PRS.

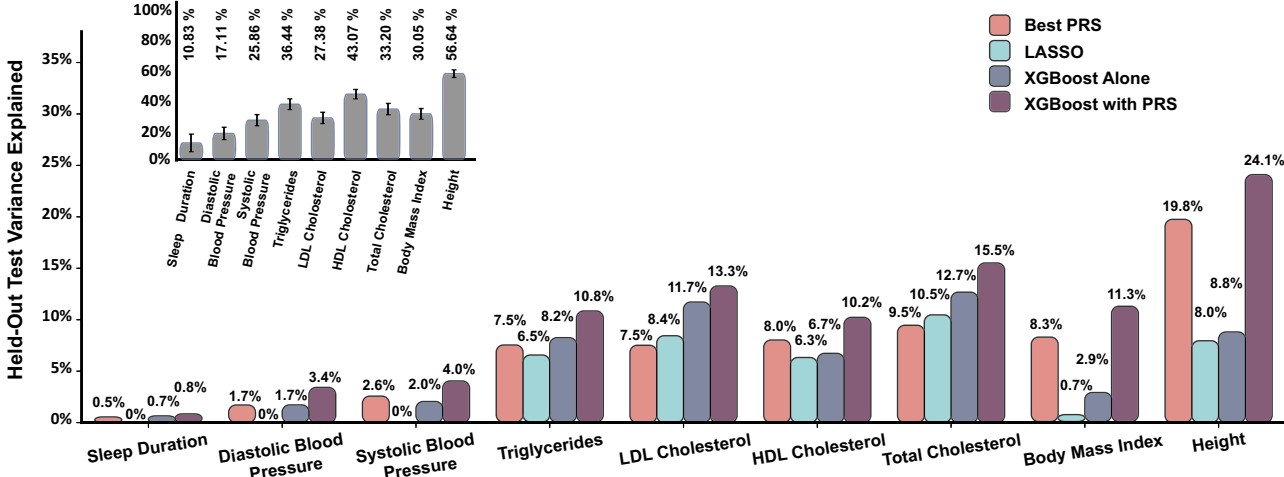

**Fig. 3 Nonlinear model consistently outperforms linear ones for prediction of multiple complex phenotypes in multi-ethnic dataset.** Linear (PRS-pink), linear-regularized (LASSO—teal), and nonlinear (XGBoost—gray, purple) models were employed to predict the harmonized phenotypes from SNP data from TOPMed following adjustment for covariates. Two versions of the XGBoost algorithm are shown with the first model employing only the SNPs as features (gray; XGBoost alone) and a second model which had the PRS as one of the features as well (XGBoost with PRS). The LASSO algorithm (teal) was trained on the same set of SNPs as the XGBoost. The inset (gray) depicts estimated heritability for same phenotypes in the same database using the REML approach with error bars of 95% confidence intervals estimated through restricted maximum-likelihood estimate.

Notably, even our best model falls short compared to the estimated heritability obtained from a linear mixed model that considers all SNPs via the kinship matrix (Fig. 3 inset). For example, we achieved ~2.3-fold better results for height (24.1% vs 56.6% PVE) and ~6.5-fold better results for systolic blood pressure (4.0% vs 25.9% PVE) with linear mixed models. These results indicate that much of the effect is distributed among a large number of weakly-correlated SNPs.

**Race/Ethnicity associates with model performance for multiple phenotypes.** Our dataset included participants with self-reported race/ethnicity (7601 Black, 14142 White, and 7320 Hispanic/Latino), with phenotype characteristics provided in Table 1. We compared the performance of the linear PRS model with the XGBoost model that includes the PRS as a feature, trained on the combined dataset, for the prediction of the different phenotypes on the ethnicity-specific datasets (Fig. 4). The hyperparameters selected for each race/ethnicity-specific model through cross-validation are listed in Supplementary Table 11, and the race/ethnicity-specific heritability is displayed in Supplementary Fig. 2.

The XGBoost with PRS model usually improves PVEs over the linear PRS model in the White and Hispanic/Latino groups, but less so in the Black group. Surprisingly, for a few phenotypes (systolic blood pressure, triglycerides, LDL cholesterol, HDL cholesterol, and total cholesterol), the PVEs were better in Hispanics/Latinos compared to Whites, even though the GWAS contained more data from Whites than Hispanic/Latino participants. However, it is important to note that many Hispanic/Latino individuals have substantial European genetic ancestry, and our study does not differentiate between Hispanic/Latinos with different levels of European genetic ancestry. Unfortunately, most models performed poorly in Black individuals.

**Ethnic diversity is crucial for model training.** Figure 5 compares XBoost models trained within race/ethnic group to the multi-ethnic model. For the Black group, the multi-ethnic model performed best on the held-out test datasets, consistently outperforming the race/ethnic-specific models – even those trained and tested on the same race/ethnic group. However, for the Hispanic/Latino group, the multi-ethnic model was sometimes inferior to the race/ethnic matched model (for sleep duration, HDL cholesterol, and body mass index). For Whites, the race/ethnic matched model improves upon the multi-ethnic model for almost all phenotypes (except LDL cholesterol, body mass index, and height). This may be due to the larger sample size available in the multi-ethnic training dataset, compared to the Black and Hispanic/Latino datasets.

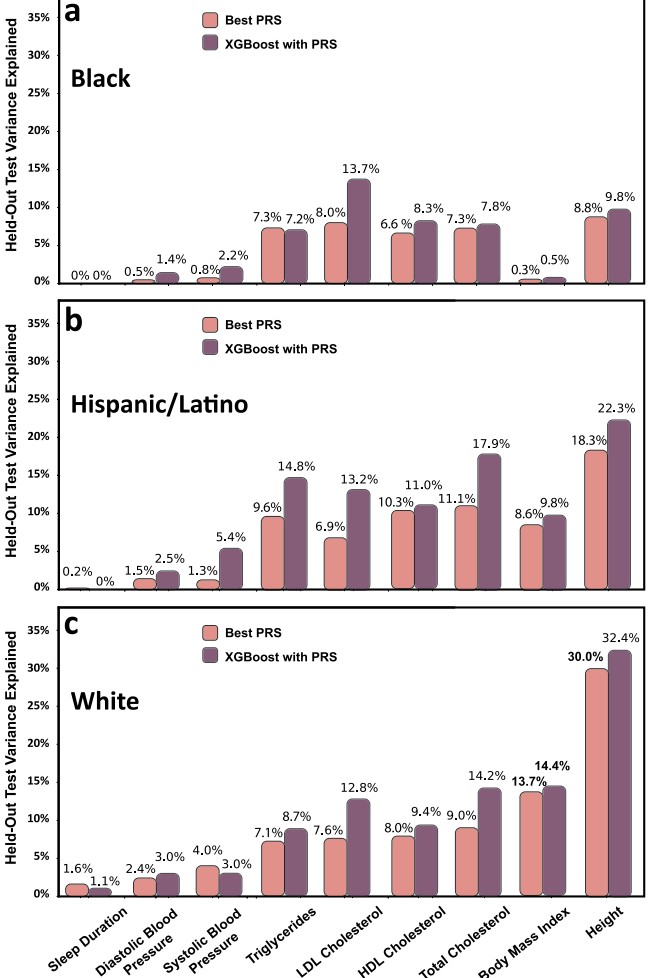

**Fig. 4 Model performance differ by group, with XGBoost consistently outperforming PRS.** Performance of the PRS (pink) and XGBoost+PRS (purple) models trained on the combined dataset when applied to the prediction of the 5 phenotypes in separate race/ethnicities. Panels **a**, **b**, and **c** refer to White, Black and Hispanic/Latino groups, respectively.

## Discussion

The aim of this study was to investigate the application of machine learning algorithms to polygenic trait prediction, specifically algorithms that allow for non-linearities and interaction effects between SNPs, and to compare their performance to linear PRS models, and LASSO methodologies that do not account for such effects. We chose nine complex phenotypes with varying levels of heritability across a large, multi-ethnic dataset including White, Black, and Hispanic/Latino participants.

Across all phenotypes in the validation dataset, we found the highest PVE by combining the XGBoost model with the PRS. The relative increase in PVE varied across the phenotypes, with up to 100% for diastolic blood pressure and 77% for LDL cholesterol. The impressive increase in the performance of the XGBoost model relative to linear PRS model and LASSO with the same SNPs as those used by the XGBoost points toward interactions between genetic alleles and/or nonlinear contributions of SNPs to phenotypes. In all cases, the XGBoost algorithm alone (without including the PRS) outperformed the linear LASSO model that used exactly the same SNPs. In half the phenotypes, however, the linear PRS performed better, likely because it could account for more weakly associated SNPs. Combining the ML model with the PRS (as a feature) achieved high prediction performance by both

accounting for the large numbers of weakly associated SNPs (linearly through PRS), in addition to some of the non-linearities and interactions (through XGBoost).

We chose to employ the XGBoost implementation of gradient boosted trees due to its strong performance in prediction tasks, explicit handling of interactions, and ability to capture nonlinear effects. The large number of potential SNPs precluded their direct inclusion into the XGBoost model, as it is extremely computationally expensive and prone to overfitting with high dimensionality. Thus, we developed an ensemble model that used the LASSO algorithm as a feature selection tool to optimize the XGBoost performance while performing a cross-validation for the hyperparameters of both LASSO and XGBoost. The inclusion of both PRS as well as XGBoost allows for direct comparison between the models, with the PRS representing the linear additive genetic contributions to the trait, allowing XGBoost to optimize the nonlinear and interaction effects.

Several studies that compared linear effects PRS models to ML models allowing for more complex genetic effects reported that linear PRS models outperform ML models. One study found that a linear Elastic Net model usually outperformed ML models that allow for nonlinear and interaction effects for prediction of gene transcripts[25]. Another study found that linear PRS models outperformed support vector machines for psychiatric phenotypes[26]. Our study differs from these prior studies by use of very large and diverse training and testing datasets (prior studies were often limited to a few thousand individuals). Our datasets also had high-quality deep sequencing and joint allele calling, as well as harmonized phenotypes across the combined dataset, which likely improved our ability to validate ML models across training and testing datasets. Also, our ensemble approach to guarding against overfitting, in addition to including the standard linear model PRS within the XGBoost model, utilized the strengths of both the linear and the nonlinear approaches in complementary ways. Specifically, this approach leveraged the ability of the PRS to capture the linear additive effects from a large number of SNPs, and the XGBoost to capture nonlinear effects and SNP-SNP interactions.

Because we compared a nonlinear ML model to a linear PRS model, we included a step where we constructed PRSs using multiple methods: clump-and threshold approach implemented using PRSice, and model-based LDPred2 and lassosum2. This is an important comparison as it is not yet clear what is an optimal approach for PRS construction when using summary statistics from GWAS based on a population with different ancestral make-up compared to the target population. PRSice-based PRSs were relatively robust to the selection of clumping parameters, however, for most traits PRSice PRSs were inferior to the best PRSs from other approaches when evaluated on the held-out test dataset. In contrast, LDPred2 performance varied substantially when using its various implementations: inf, auto, and grid. LDPred2-auto had better performance than its counterparts. Possible explanations are that the grid implementation overfitted (the best-performing parameter combination in the training dataset may not have been ideal for the test dataset), and that the inf model is misspecified. Lassosum2 tended to have superior performance compared to other PRSs. We note that while lassosum2 approximates a LASSO regression, the results when using lassosum2 are different than the results when using LASSO. This is likely due to two reasons, First, the selection of SNPs used: because standard LASSO implementation cannot handle, computationally, too many SNPs, we implemented it using SNPs with $p$ value $< 10^{-4}$ and further divided into five sets of SNPs. In contrast, lassosum was implemented using 1 M SNPs with the lowest $p$ values in the summary statistics (and that were also available in our dataset),

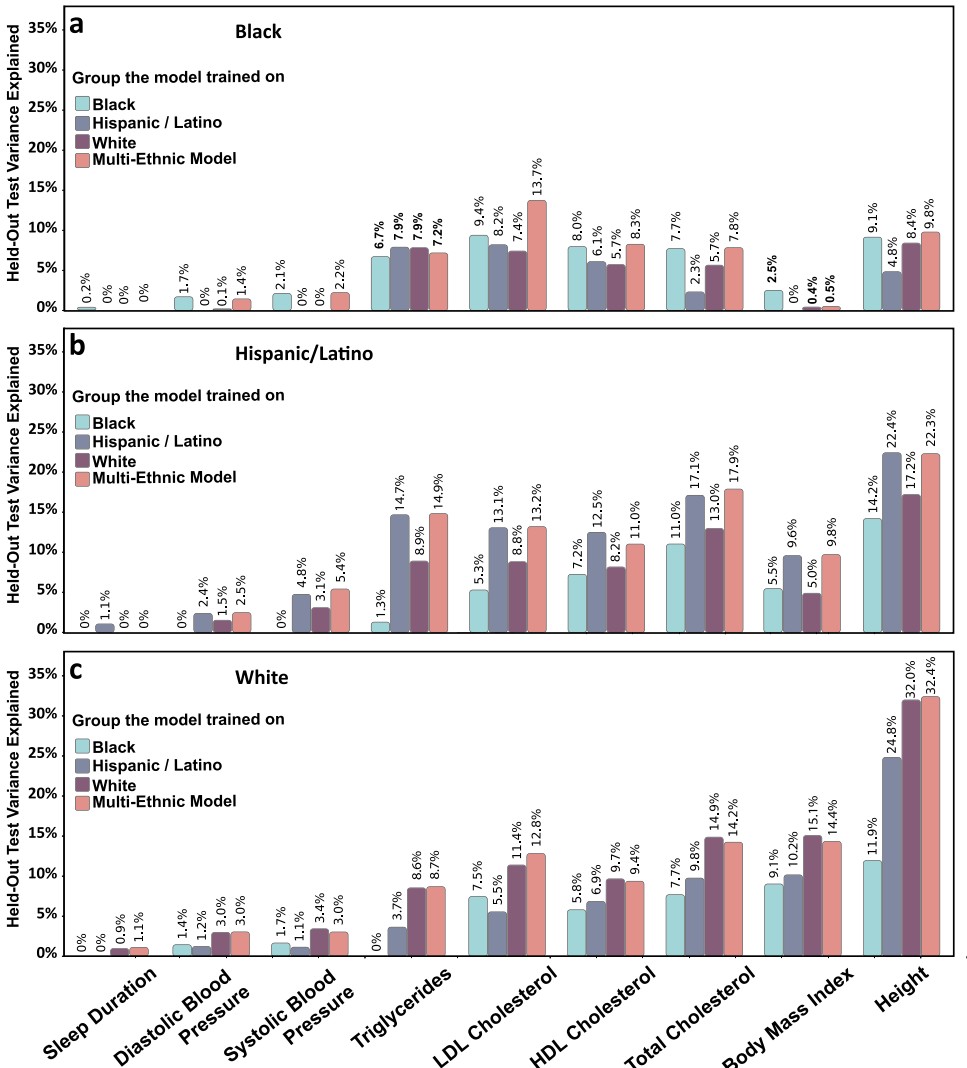

**Fig. 5 Multi-ethnic XGBoost model performs on par with the race/ethnic-specific models.** XGBoost with PRS models were trained either on the combined dataset containing all participants, (pink) or on each race/ethnic group separately (teal, gray and purple). The models were then evaluated on each of the groups (**a** Black, **b** Hispanic/Latino, and **c** White).

without clumping. Second, the lassosum model assumes that the marginal SNP effect sizes are as supplied by the GWAS summary statistics, while the LASSO model does not have such an assumption and it only relies on the available individual-level data.

We compared multi-ethnic and race/ethnicity-specific models and found that multi-ethnic-trained models on large datasets had different performance across race/ethnic groups. In our analyses, the multi-ethnic models had better performance in the White and Hispanic/Latino groups than in the Black group. Models trained using the same race/ethnic group and the multi-ethnic trained model had similar prediction performance, despite a substantial decrease in training sample size in the multi-ethnic model. Overall, the PVE for the studied phenotypes was consistently lower for Black participants than for White or Hispanic/Latino participants. The difference in PVE varied by phenotype, usually between 1.3–2.1 times lower for Black participants compared to White participants. There are several possible explanations for these findings. First it may be that the combined models predominately use European ancestry-specific genetic effects. Both the White and the Hispanic/Latino groups have substantial European ancestry, while Black individuals have lower European ancestry. Specifically, across Hispanic/Latino

background groups reported in the Hispanic Community Health Study/Study of Latinos, on average 40-80% have European ancestry[27] while in the Jackson Heart Study, Black individuals are estimated to have 16% European ancestry on average[28]. Where possible, we used multi-ethnic GWAS analyses to select candidate SNPs for analysis. However, most GWAS participants are still White[29]. Therefore, the choice of SNPs is more optimal for groups with substantial amount of European ancestry, so that SNPs with small effects or low minor allele frequencies (MAF) in European ancestry populations and larger effects or higher MAF in African ancestry populations were not discovered in the GWAS and were therefore not selected to be used in the trained prediction models. This limitation has been shown to reduce PRS performance in African and African Americans populations in multiple studies[30–32].

There are some limitations to this study. First, while the TOPMed cohort is diverse, White participants are over-represented. Second, as noted above, although the GWAS analyses that we relied upon were multi-ethnic (other than sleep duration and height GWAS which were based European ancestry samples), it seems likely that important variants for these phenotypes among a Black population do not achieve the required $p$ value level ($<10^{-4}$) to be included, given limited sample sizes

for Black participants in these prior GWAS analyses. Similarly, we have not considered the interactions between non-GWAS-selected SNPs, which may lead to some important variants with interaction effects being excluded. Third, much of our ensemble algorithm relies on feature selection. This may be overly restrictive and does not allow for variants with very small effect sizes to be included (as noted in the results for Height). It is also possible that SNPs selected through LASSO may not be prioritized based on nonlinear or interaction effects, even though we model them using the nonlinear XGBoost. A promising area of future research could be using XGBoost on the full set of candidate SNPs to perform feature selection, and then use LASSO (or another algorithm) for prediction of classification, while potentially including interaction terms and other SNP models (dominant, recessive) as features. A limitation of our current computational infrastructure and the size of the dataset is the inability to run XGBoost on many hundreds of thousands of SNPs, which, if ameliorated, would allow us to use XGBoost for feature selection, as some other studies have done[16–18]. Fourth, we used self-reported race/ethnicity. An alternative grouping would use genetically determined ancestry groups. We chose self-reported grouping to better approximate clinical settings and to potentially account for gene-environment interactions, in which people who share self-reported race/ethnicity may have more similar environmental exposures, compared to individuals outside the group. Fifth, we use ML as a tool to model the interactions and non-linearities. However, this approach does not explicitly identify individual interactions or non-linearities nor quantifies the contributions of each.

Finally, due to the high complexity of the XGBoost model and pre-filtering of SNPs, highly polygenic traits (such as Height) suffer in performance when compared to less complex phenotypes. The XGBoost models performed less well in traits that had a large number of candidate SNPs selected by LASSO, likely due to overfitting. A potential approach to address this is to force the ensemble model to select less SNPs into the XGBoost model by applying LASSO over a smaller set of SNPs by imposing a stricter $p$ value threshold on the SNPs provided to the LASSO step, or by considering a narrower range of potential penalization parameter values for LASSO, corresponding to less selected SNPs. It is a topic of future work to assess such approaches, and to evaluate the potential trade-off, ideally in simulation studies, between the included proportion of potential SNPs used in the prediction model and prediction accuracy, while accounting for potential overfitting.

Overall, this study uncovers strong evidence for contributions of nonlinear genetic effects and interaction between alleles to complex phenotypes. Additionally, our findings re-iterate one of the largest hurdles for better performing, robust genetic prediction models across diverse individuals—namely the lack of well-powered GWAS for different race/ethnic groups and subpopulations[33]. This work opens up promising avenues for future research, such as: creating a generalizable tool that would allow ML PRS to be deployed on other studies; estimating the individual contributions of interactions and non-linearities; and developing approaches to prioritize SNPs for inclusion in the ML model that would increase predictive ability in Black and other non-White populations.

## Methods

**Study population.** The study sample included 34,072 unrelated (3rd degree or less) TOPMed participants from eight U.S. based cohort studies: Jackson Heart Study (JHS; $n = 2504$), Framingham Heart Study (FHS; $n = 3520$), Hispanic Community Health Study/Study of Latinos (HCHS/SOL; $n = 6,408$), Atherosclerosis Risk in Communities study (ARIC; $n = 6197$), Cardiovascular Health Study (CHS; n = 2835), Multi-Ethnic Study of Atherosclerosis (MESA; $n = 3949$), Cleveland Family Study (CFS; $n = 1182$), and Coronary Artery Risk

Development in Young Adults Study (CARDIA; $n = 2468$). Study descriptions are provided in Supplementary Note 1. Phenotypes were harmonized by the TOPMed Data Coordinating Center (DCC)[34], and included age, sex, race/ethnicity, study (used as covariates), phenotypes of interest, and medications, which were used to adjust measures of relevant phenotypes (Supplementary Table 12). Note that sex was self-reported and verified by chromosomal sex, and therefore biological sex and gender identify in these analyses are the same. Thus, we refer to this variable as "sex". The dataset included 7601 non-Hispanic Black participants, 14,142 non-Hispanic White participants, and 7320 participants of Hispanic/Latino descent. The dataset was divided such that 20% of the data was held out as a validation set. A secondary analysis used a larger training dataset that included related individuals but in which all individuals were still unrelated to those in the test dataset. Sex-stratified analyses were not performed due to limited sample size.

**Ethical regulations.** Participants from each of the studies contributing to the TOPMed consortium provided informed consent, and all studies were approved by IRBs in each of the participating institutions. In detail, the HCHS/SOL was approved by the institutional review boards (IRBs) at each field center, where all participants gave written informed consent, and by the Non-Biomedical IRB at the University of North Carolina at Chapel Hill, to the HCHS/SOL Data Coordinating Center. All IRBs approving the study are Non-Biomedical IRB at the University of North Carolina at Chapel Hill. Chapel Hill, NC; Einstein IRB at the Albert Einstein College of Medicine of Yeshiva University. Bronx, NY; IRB at Office for the Protection of Research Subjects (OPRS), University of Illinois at Chicago. Chicago, IL; Human Subject Research Office, University of Miami. Miami, FL; Institutional Review Board of San Diego State University. San Diego, CA. The Framingham Heart Study was approved by the Institutional Review Board of the Boston University Medical Center. All study participants provided written informed consent. The ARIC study has been approved by Institutional Review Boards (IRB) at all participating institutions: University of North Carolina at Chapel Hill IRB, Johns Hopkins University IRB, University of Minnesota IRB, and University of Mississippi Medical Center IRB. Study participants provided written informed consent at all study visits. All CHS participants provided informed consent, and the study was approved by the Institutional Review Board of the University Washington. All MESA participants provided written informed consent, and the study was approved by the Institutional Review Boards at The Lundquist Institute (formerly Los Angeles BioMedical Research Institute) at Harbor-UCLA Medical Center, University of Washington, Wake Forest School of Medicine, Northwestern University, University of Minnesota, Columbia University, and Johns Hopkins University. All CARDIA participants provided informed consent, and the study was approved by the Institutional Review Boards of the University of Alabama at Birmingham and the University of Texas Health Science Center at Houston. The JHS study was approved by Jackson State University, Tougaloo College, and the University of Mississippi Medical Center IRBs, and all participants provided written informed consent. The Cleveland Family Study was approved by the Institutional Review Board (IRB) of Case Western Reserve University and Mass General Brigham (formerly Partners HealthCare). Written informed consent was obtained from all participants.

**Genotype data.** We used whole genome-sequencing data from TOPMed[35] Freeze 8, without restriction on sequencing depth, which contains 705,486,649 variants. The dataset includes samples sequenced through the National Human Genome Research Institute's Centers for Common Disease Genomics (CCDG) program, where the sequence data for all TOPMed and CCDG samples were harmonized together via joint allele calling. The methods for TOPMed WGS data acquisition and quality control (QC) are provided in https://www.nhlbiwgs.org/topmed-whole-genome-sequencing-methods-freeze-8. Principal Components (PCs) and kinship coefficients were computed for the genetic data by the TOPMed DCC using the PC-Relate algorithm[36] implemented in the GENESIS R package[37]. In this work, we used 5 PCs computed via the GENESIS R package PC-Air algorithm[38] to adjust for global ancestry. Based on the kinship coefficients, we identified related individuals and generated a dataset in which all individuals were degree-3 unrelated, i.e., all kinship coefficients were lower than 0.0625. We extracted allele counts of variants that passed QC from GDS files using the SeqArray[39] package version 1.28.1 and then further processed using R and Python scripts. After QC and filtering variants with MAF < 0.01 (with MAF being computed based on the multi-ethnic TOPMed dataset), we had 12,482,699 variants in the TOPMed data. For all variants, we set the effect allele to be the minor allele.

**Heritability estimation.** Let $\mathbf{K}$ denote an $n \times n$ kinship matrix, having twice the kinship coefficient between the $i$th and $j$th participants in its $I, j$ entry. For an outcome $y$, we assume the linear model:

$$y_i = X_i \beta + \epsilon_i,$$
$$cov(\boldsymbol{\epsilon}) = \sigma_e^2 \boldsymbol{I}_{n \times n} + \sigma_k^2 \boldsymbol{K} \qquad (1)$$

where $\boldsymbol{\epsilon} = (\epsilon_1, \ldots, \epsilon_n)^T$ is the normally-distributed vector of errors across the sample. We estimated the narrow-sense heritability $\hat{h}^2 = \hat{\sigma}_k^2 / \hat{\sigma}_k^2 + \hat{\sigma}_e^2$ using the

**Table 3 Description of published GWAS used to identify summary statistics.**

| Phenotype | GWAS | Study population | Participants |
|---|---|---|---|
| Height, BMI | Meta-analysis of genome-wide association studies for height and body mass index in ~700,000 individuals of European ancestry[54] | UK Biobank and the GIANT consortium | 693,529 (European ancestry) |
| Total cholesterol, LDL cholesterol, HDL cholesterol, triglycerides | Genetics of blood lipids among ~300,000 Multi-ethnic Participants of the Million Veteran Program[55] | Million Veteran Program | 297,626 (72.4% non-Hispanic Whites, 19.3% non-Hispanic Blacks, 8.3% Hispanics) |
| Systolic blood pressure, diastolic blood pressure | Trans-ethnic association study of blood pressure determinants in over 750,000 individuals[56] | Million Veteran Program | 318,891 (69.1% non-Hispanic Whites, 18.8% non-Hispanic Blacks, 6.7% Hispanics, 0.77% non-Hispanic Asians and 0.85% non-Hispanic Native Americans) |
| Sleep duration | Genome-wide association study identifies genetic loci for self-reported habitual sleep duration supported by accelerometer-derived estimates[57] | UK Biobank | 446,118 (European ancestry) |

GWAS source, study population as reported by the manuscript reporting the GWAS, and number of participants used to generate summary statistics for a given phenotype.

Restricted Maximum-Likelihood approach as implemented in the GCTA[40] software (version 1.93.2). Confidence intervals at the 95% level are provided using standard errors (SEs) approach and based on the assumption of an asymptotic normal distribution (estimate $\pm 1.96 \times$ SE).

**Phenotypes.** We trained genetic prediction models to predict sleep duration, diastolic blood pressure, systolic blood pressure, triglycerides, LDL cholesterol, HDL cholesterol, total cholesterol, body mass index, and height; and we used sex, study, race/ethnicity, and age as covariates. For reproducibility, Supplementary Table 12 provides the coded names of each of the phenotypes and covariates used in the analysis and describes in detail transformations and exclusions.

For each of the phenotypes of interest, we excluded outlying individuals defined by phenotypic values above the 99th quantile and values below the 1st quantile for the phenotype, computed over the multi-ethnic dataset. Values of systolic blood pressure in individuals using antihypertensive medications were set to missing, and similarly, values of triglycerides and total cholesterol levels of individuals using cholesterol medications. Triglycerides and total cholesterol values were log transformed to obtain an approximate normal distribution. Values of diastolic blood pressure in individuals using antihypertensive medications were increased by 10 mmHg. Then, each phenotype was regressed on age, sex, study, and race/ethnicity. The residuals were extracted and rank-normalized. Subsequent analyses used these rank-normalized residuals as the outcomes[41], and we refer to them henceforth as adjusted phenotypes.

**Summary statistics from published GWAS.** We used summary statistics from published GWAS to select SNPs and their weights to construct PRS, as well as to select SNPs to include in the ML models. The GWAS used for each of the nine phenotypes are described in Table 3. When possible, we used multi-ethnic GWAS. We lifted over the coordinates to our genome build GRCh38/hg38 using the LiftOver tool from the UCSC genome browser[42]. For about half of the phenotypes, 90% or more of the variants in the GWAS were found in the TOPMed data. For the other half, 60–70% were found (Supplementary Table 13).

**Polygenic risk score (PRSice, LDpred2, and lassosum2).** We calculated the standard PRS using the classic clump-and-threshold methodology (PRSice PRS)[2]. We used PRSice 2 software version 2.3.1[43] to calculate the genetic score. SNPs with ambiguous alleles were removed using the PRSice software. We calculated PRS using two clumping regions (250 and 500 kb) on each side of the index SNP and three clumping $R^2$ values (0.1, 0.2, 0.3). We considered $p$ value thresholds of 0.5 through $1e^{-10}$. For each adjusted phenotype and each PRS defined by clumping region, clumping $R^2$ value, and $p$ value threshold, we fit a linear model including covariates, the PRS, and genetic PCs to account for population structure[44]. We selected the PRS where the PRS model minimized the mean squared error in the training dataset. We assessed the percentage of variance explained (PVE) by the PRS models using the methodology described below.

We also calculated the PRS using LDpred2[45] (LDpred2 PRS). We used the entire multi-ethnic TOPMed data to perform clumping with PRSice and to compute SNP weights using LDpred2. We used R package bigsnpr version 1.9.10 to calculate the genetic score with LDpred2-inf, which utilizes the infinitesimal model, LDpred2-grid which uses a grid of values for the hyperparameters, and LDpred2-auto, which is an automatic estimation of the proportion of causal variants ($p$) and the SNP heritability ($h^2$) from the data, without any tunable hyperparameters. Because LDpred2 can compute joint weights for up to ~1 million SNPs, we used

two approaches to select SNPs: (a) we pruned SNPs using pruning parameters $R^2 = 0.1$ and distance $= 500$ kb; and (b) the top 1 million SNPs with respect to their association $p$ value in the summary statistics (and that overlapped with the TOPMed dataset) to train the model. Following the recommendation provided by LDpred2 manuscript, we used the bigsnpr R package to correct the estimated effect sizes for the winner's curse prior to applying LDpred2.

Finally, we calculated PRS via penalized regression on summary statistics, lassosum2[46]. We used the lassosum2 implementation in R packages bigsnpr v1.9.5 and bigstatsr v1.5.6[47], using the default hyperparameters as described in detail in Privé et al.[48], including the top 1 million SNPs from each GWAS.

All methods (PRSice, LDpred2, and lassosum2) require reference panel for LD inference used for clumping (PRSice) or for tuning SNP effect sizes (LDpred2, lassosum2). We used the multi-ethnic TOPMed dataset as an LD reference panel.

After calculating the C+T PRSice PRS, LDpred2 PRS, and lassosum2 PRS, we selected the best-performing PRS in the training set for the purposes of comparison between linear PRS models and XGBoost.

**LASSO and Gradient Boosted Trees (XGBoost) ensemble.** Figure 2 describes the construction of an ensemble ML model for polygenic risk prediction. We considered for inclusion in the models all SNPs having $p$ value $< 1 \times 10^{-4}$ in the corresponding GWAS and used them to develop an ensemble prediction model. In brief, the ensemble model included two steps: (1) a LASSO penalized regression for filtering candidate SNPs; and (2) an XGBoost prediction model allowing for nonlinear interactions. In detail, gradient boosted trees are a widely used machine learning technique that creates an ensemble of weak decision trees (i.e., limited in depth or interactions) by iteratively optimizing an objective function at each boosting step in which new trees are optimized based on the residuals of the previous boosting step. XGBoost is an optimized implementation of gradient boosted trees that is highly efficient in distributed computing environments[13]. However, the set of candidate SNPs is very large for most of the GWAS listed in Table 3, and boosting is prone to overfitting with high dimensionality[49]. LASSO[50] is a commonly used model for feature selection that can mitigate overfitting by encouraging parsimony through L1 regularization. We trained an ensemble model, jointly training LASSO and XGBoost models in order to prevent overfitting due to the high dimensionality of genetic data (through LASSO) while simultaneously exploiting the nonlinear relationships and interaction effects (through XGBoost).

The ensemble model was trained as follows. For each given regularization hyperparameter $\alpha \in \{0 \dots 1\}$ we fit LASSO on the training dataset using a 10-fold cross-validation scheme (and the MSE loss). The LASSO model included linear SNP effects and unpenalized covariates and, to reduce required computational resources, it was separately fit using the same $\alpha$ on 5 sets of SNPs, each including all SNPs from a few chromosomes, set so that the number of SNPs is roughly equivalent between models. Next, we filtered to SNPs with non-zero coefficients in the LASSO model. Using these selected SNPs, we fit the XGBoost model via 3-fold cross-validation applied on the training dataset, allowing up to 10,000 boosted trees with early stopping after 10 rounds of boosting without improvement in the threefold cross-validation loss (see Table 2 for details). Based on this threefold cross validation, we selected the number of trees $\theta_\alpha$ that minimized the mean squared prediction error (MSE), resulting in a set of parameters $(\alpha, \theta_\alpha)$. We selected the optimal $(\alpha, \theta_\alpha)$ pair that minimized the MSE of the threefold cross-validation step across all values of $\alpha$. For XGBoost, we always used a learning rate of 0.01, maximum depth of 5, column sample by tree of 90%, minimum child weight of 10, and subsample of 50%. Finally, we performed LASSO regression using the same variants that were selected in this process, to explicitly compare the results of a nonlinear model allowing for interactions to a linear model without interactions.

We performed this process individually for each of our nine adjusted phenotypes using a distributed cluster computing environment. This was a regression task. All models included the ancestral PCs, and some models used the best-performing PRS as a variable. We assessed the PVE of the genetic machine learning models using the methodology described below and compared the results with the best-performing linear PRS model. Analysis was conducted using Python 3 and the *scikit-learn*[51] and *xgboost* packages[13].

**Secondary analysis comparing the ensemble model with a standard linear PRS model using the same potential SNP set.** Because we limited the SNPs used by the ensemble model to those with $p$ value $< 10^{-4}$ in their discovery GWAS, in a secondary analysis we compare the performance of the ensemble models to linear PRS models with C+T PRS that use the $p$ value $< 10^{-4}$ threshold for SNP selection. Thus, the two models rely on the same set of candidate SNPs.

**Secondary analyses studying SNP selection into the XGBoost model.** We performed two experiments to test the benefit of using LASSO to select SNPs prior to the XGBoost model. Each experiment considered a different way to select SNPs into the XGBoost model. In the first experiment, we selected SNPs at random as a baseline for four phenotypes (total cholesterol, triglycerides, LDL cholesterol, and HDL cholesterol). We used the same number of SNPs as the number selected by LASSO in the respective XGBoost Alone and XGBoost with PRS models. We have performed this experiment for four phenotypes (total cholesterol, triglycerides, LDL cholesterol, and HDL cholesterol), by (1) randomly selecting SNPs in the same size as the LASSO selected SNPs for those phenotypes, 92) running the XGBoost model with and without PRS, (3) repeating 100 times, and (4) averaging the result. In the second experiment, for LDL cholesterol we used the SNPs with non-zero weighting in the lassosum2 PRS as the selected SNPs in the XGBoost model. We used a limited set of phenotypes for these experiments due to computational limitations.

**Race/ethnicity analysis.** We first trained the models using the combined, multi-ethnic dataset (multi-ethnic model). We then trained the models on the subset of the sample containing only White, Black, and Hispanic/Latino participants. This resulted in four models that were each trained on different race/ethnicity groups: Multi-Ethnic, White, Black, and Hispanic/Latino. For each of these four models, we assessed the PVE among the participants of each race/ethnicity in the held-out test set.

**Model evaluation in the held-out test set.** We quantify model performance as the variance explained (sometimes referred to as the adjusted $R^2$). Let $y_i^0, i = 1, \ldots, n$ denote the adjusted phenotype. $\text{Var}(y^0)$ estimates the total baseline model variance. For a given model $m$, let $\hat{y}_i^m$ denote the predicted (adjusted) phenotype value for the $i$th person. We estimate the percent variance explained by model $m$ as:

$$\text{PVE} = \left(1 - \frac{\text{var}(y^0 - \hat{y}^m)}{\text{var}(y^0)}\right) \times 100\%. \quad (2)$$

We compute the relative PVEs between various models as the relative percentage increase, i.e., $(\text{PVE}_2 - \text{PVE}_1)/\text{PVE}_1$.

**Reporting summary.** Further information on research design is available in the Nature Research Reporting Summary linked to this article.

## Data availability
TOPMed freeze 8 WGS data are available by application to dbGaP according to the study-specific accessions: FHS: phs000974.v4.p3, JHS: phs000964.v1.p1, MESA: phs001211.v3.p2, CARDIA: phs001612.v1.p1, CFS: phs000954.v3.p2, CHS: phs001368.v2.p1, HCHS/SOL: phs001395.v1.p1, ARIC phs001416.v2.p1. Study phenotypes are available from dbGaP from parent studies accession: FHS: phs000007.v32.p13, JHS: phs000286.v6.p2, MESA: phs000209.v13.p3, CARDIA: phs000285.v3.p2, CFS: phs000284.v2.p1, CHS: phs000287.v7.p1, HCHS/SOL: phs000810.v1.p1, ARIC: phs000090.v7.p1. Instructions to generate PRS that were used in this manuscript, i.e., SNP identifiers (chromosome and positions in genome build hg38) and alleles are publicly available in a figshare repository[52] https://doi.org/10.6084/m9.figshare.20304135.v1. Ensemble ML models for each of the phenotypes trained over both multi-ethnic and race/ethnic groups are publicly available in a figshare repository[53] https://doi.org/10.6084/m9.figshare.20301423.v1. Supplementary Data 1 provides the source data behind the figures in the manuscript and behind the supplementary figures.

## Code availability
Code used for the analyses in this manuscript is provided on a dedicated GitHub repository https://github.com/genevievelyons/MachineLearning_PolygenicRiskScore, https://doi.org/10.5281/zenodo.6964364. We used SeqArray[39] package version 1.28.1,

GCTA[40] software version 1.93.2, PRSice 2 software version 2.3.1[43], R package bigsnpr version 1.9.10, Python 3 and the *scikit-learn*[51] and *xgboost* packages[13].

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

## Acknowledgements

Molecular data for the Trans-Omics in Precision Medicine (TOPMed) program was supported by the National Heart, Lung and Blood Institute (NHLBI). Genome sequencing for "NHLBI TOPMed: Whole Genome Sequencing and Related Phenotypes in the Framingham Heart Study" (phs000974.v4.p3) was performed at the Broad Institute Genomics Platform (3R01HL092577-06S1, 3U54HG003067-12S2). Genome sequencing for "NHLBI TOPMed: The Jackson Heart Study" (phs000964.v1.p1) was performed at the Northwest Genomics Center (HHSN268201100037C). Genome sequencing for the "NHLBI TOPMed: The Atherosclerosis Risk in Communities Study" (phs001211.v3.p2) was performed at the Broad Institute Genomics Platform (3R01HL092577-06S1) and the Baylor College of Medicine Human Genome Sequencing Center (HHSN268201500015C, 3U54HG003273-12S2). Genome sequencing for "NHLBI TOPMed: Coronary Artery Risk Development in Young Adults Study" (phs001612.v1.p1) was performed at the Baylor College of Medicine Human Genome Sequencing Center (HHSN268201600033I). Genome sequencing for "NHLBI TOPMed: Cleveland Family Study" (phs000954.v3.p2) was performed at the Northwest Genomics Center (3R01HL098433-05S1, HHSN268201600032I). Genomics sequencing for "NHLBI TOPMed: Cardiovascular Health Study" (phs001368.v2.p1) was performed at the Baylor College of Medicine Human Genome Sequencing Center (3U54HG003273-12S2, HHSN268201500015C, HHSN268201600033I). Genome sequencing for "NHLBI TOPMed: Hispanic Community Health Study/Study of Latinos" (phs001395.v1.p1) was performed at the Baylor College of Medicine Human Genome Sequencing Center (HHSN268201600033I). Genome sequencing for "NHLBI TOPMed: Multi-Ethnic Study of Atherosclerosis" (phs001416.v2.p1) was performed at Broad Institute Genomics Platform (HHSN268201500014C, 3U54HG003067-13S1). Core support including centralized genomic read mapping and genotype calling, along with variant quality metrics and filtering were provided by the TOPMed Informatics Research Center (3R01HL-117626-02S1; contract HHSN268201800002I). Core support including phenotype harmonization, data management, sample-identity QC, and general program coordination were provided by the TOPMed Data Coordinating Center (R01HL-120393; U01HL-120393; contract HHSN268201800001I). We gratefully acknowledge the studies and participants who provided biological samples and data for TOPMed. The Genome Sequencing Program (GSP) was funded by the National Human Genome Research Institute (NHGRI), the National Heart, Lung, and Blood Institute (NHLBI), and the National Eye Institute (NEI). The GSP Coordinating Center (U24 HG008956) contributed to cross-program scientific initiatives and provided logistical and general study coordination. The Centers for Common Disease Genomics (CCDG) program was supported by NHGRI and NHLBI, and whole genome-sequencing was performed at the Baylor College of Medicine Human Genome Sequencing Center (UM1 HG008898 and R01HL059367). M.E., T.S., and S.R. are supported by NHLBI grants R35HL135818 to SR. G.M.P. is supported by R01HL142711 from the NHLBI. The project described was supported by the National Center for Advancing Translational Sciences, National Institutes of Health, through grant KL2TR002490 to L.M.R. P.d.V. was supported by American Heart Association grant number 18CDA34110116 and NHLBI grant number R01HL146860. The views expressed in this manuscript are those of the authors and do not necessarily represent the views of the National Heart, Lung, and Blood Institute; the National Institutes of Health; or the U.S. Department of Health and Human Services.

## Author contributions

M.E., G.L., S.R.B., and N.K. constructed PRS. M.E. and G.L. developed ML models, performed association analyses, and summarized results in tables and figures. M.E., G.L., and T.S. conceptualized and drafted the manuscript. TS supervised the work for this manuscript. J.A.B., X.G., L.R., Y.G., L.A.L., G.M.P., J.I.R., S.S.R., A.C.M., B.M.P., D.M.L., and S.R. designed data collection and/or TOPMed sample selection in the cohort they represent and designed best-practices for data analysis for the same cohorts. S.R.B., N.K., J.A.B., X.G., H.J.L., L.R., Y.G., H.C., P.V., D.M.L., L.A.L., G.M.P., M.F., J.I.R., S.S.R., A.C.M., B.M.P., D.L., and S.R. critically reviewed the manuscript.

## Competing interests

The authors declare the following competing interests: B.M.P. serves on the Steering Committee of the Yale Open Data Access Project funded by Johnson & Johnson. G.L. is a full-time employee of Valo Health, a technology company, but this work was not conducted in that position and is not relevant to that position. All other co-authors declare no competing interests.

**Additional information**

## the NHLBI's Trans-Omics in Precision Medicine (TOPMed) Consortium

Jennifer A. Brody [5], Xiuqing Guo [6], Henry J. Lin [6], Laura Raffield [7], Yan Gao[8], Paul de Vries[9], Donald M. Lloyd-Jones[11], Leslie A. Lange[12], Gina M. Peloso [13], Myriam Fornage [9,14], Jerome I. Rotter[6], Stephen S. Rich [15], Bruce M. Psaty [16], Daniel Levy[17,18], Susan Redline [1,2] & Tamar Sofer [1,2,3 ✉]

