## [Peer Review File · Communications Biology]

Reviewers' comments:

Reviewer #1 (Remarks to the Author):

Elgart et al. present an interesting study on polygenic risk scores where they find that gradient boosted trees capture non-linear effects and epistatic interactions. I personally believe that the research question raised is important as it can improve the accuracy of polygenic risk scores and their value in studying genetics and in clinical settings. I also found the paper to be well written and I appreciate the effort made to include multi-ethnic data. However, I would have liked to see a greater comparison of methods, as I believe the benefit seen by gradient boosted trees could be partially due to the few number of variants included in the PRSs, and how they are generated. I therefore have a number of comments that I would like to see addressed in a revised version.

1. Clumping and thresholding (C+T), as implemented in PRSice is as the authors note still a common strategy. However recently multiple methods have been proposed that seem to consistently outperform C+T in benchmark studies (Ni et al., *Biol PSych* 2021, <https://doi.org/10.1016/j.biopsych.2021.04.018>; Kulm et al., *medRxiv* 2021, <https://doi.org/10.1101/2020.04.06.20055574>; Pain et al., *PLoS Genet* 2021, <https://doi.org/10.1371/journal.pgen.1009021>; Zhou and Zhao, *PLoS Genet* 2021, <https://doi.org/10.1371/journal.pgen.1009697>). I would like to see comparisons with methods like PRS-CS, LDpred2, SBayesR, SDPR, and lassosum (at least one such method). I am curious to see whether the improvement observed here is also observed when using these instead of C+T.

2. The authors are absolutely correct that C+T will generally miss many haplotypes and epistatic effects. However, if we believe some of these haplotypes and epistatic effects are captured by neighbouring variation, it is possible that including more variants in the polygenic score can improve the prediction. I am worried that this might in particular be a problem for C+T due to the LD-clumping step, which seems to be chosen to be very strict ($r^2=0.1$). This results in very few variants being actually used when, e.g., predicting height. Hence, I recommend trying different LD clumping parameters, e.g. $r^2=0.2$ and a larger LD window.

2.b. Maybe the strict thresholding applied might lead to a more robust PRS with respect to predicting into multi-ethnic data. That is something one can actually try out on real data, i.e. are the new scores more or less robust (wrt genetic ancestry) when training using individuals of one genetic ancestry and predicting into a sample with a different genetic ancestry.

3. Regarding citation 16, they report in my opinion a suspiciously high AUC for CAD PRS. Alternative citations pointing out the value of capturing non-linear effects could include Sigurdsson et al., (*bioRxiv* 2021, <https://doi.org/10.1101/2021.06.11.447883>), but they find that NNs can improve prediction over lasso for immune-related diseases, but importantly, not all diseases.

4. In line 103-104 you say "we hypothesize that a large and ancestry diverse cohort would improve genetic prediction across populations". Although I intuitively agree with this, I believe the proof is in the pudding. More specifically, I would like to see whether this is actually the case for the analysis proposed here, or maybe citations that do examine this in details, e.g. by trying different approaches.

5. You restrict the analysis to 5 complex outcomes. Why not more, so that the statement could become more general. In particular, I would be interested in immune-related diseases. I would also like to see whether your results hold in UKB data, if possible. The UKB would also enable you to have "large" training data.

6. It is unclear to me how much overlap there was between the variants available in the GWAS sum stats and the 705M variants in the TOPMed data. Also, what LD reference did you use for clumping in PRSice? I am worried that using multi-ethnic GWAS sum stats can create problems when adjusting for

LD in the LD-clumping step. The LD reference should ideally reflect the LD in the GWAS sample, which in practice means that the ancestry composition of the GWAS sample should be similar to the one in the LD reference sample. I would appreciate details on this as it will impact the accuracies of the polygenic scores. (It would of course be best if LD references were always released together with the GWAS sum stats, but that's currently not the case.)

6.b Also, what additional filters did you apply, e.g. MAFs, etc.

6.c How did you deal with variants with ambiguous alleles, C/G and A/T, when matching variants between the GWAS sum stats and the TOPMed data?

7. PRSice by default tries extremely many thresholds, which I have found to lead to overfitting problems in some instances. To address this, one can either reduce the number of thresholds tested, e.g. try only 10-20 thresholds, or one can use some methods that do not require validation data, for fitting hyperparameters, e.g. SBayesR, PRS-CS-auto, LDpred2-auto, etc.

8. Equation in line 241 is sometimes referred to as adjusted R².

Reviewer #2 (Remarks to the Author):

The main claim of the study is that combining linear and non-linear machine learning models better works for phenotype predictions based on genetic data:

- How SNPs are encoded before inputting to the models? Have you tested changing the encoding scheme and look at results again? You may get different sets of potential SNPs to input to the models.

- Can authors justify why adding a single feature as PRS to a stochastic model as XGBoost considerably increases the PVE values?
Can you demonstrate the distribution of prediction scores of the classes before and after adding PRS to the XGBoost model?

- One of the draw backs of the study is that the authors still consider known SNPs that have been identified by GWAS, and as they have also mentioned GWAS SNPs have been computed one-by-one without considering interaction between them. From the material, authors have over 7m SNPs. How do you consider the interaction between non-GWAS selected SNPs and generally unknown SNPs?

- What is the loss function for XGBoost and Lasso? Is the task classification or regression? If classification, why you considered MSE loss, that is for regression tasks?

- Can you repeat an experiment with randomly selecting SNPs same size as the Lasso selected SNPs and then add PRS feature to it? Please repeat this randomized experiment for 100-1000 times and average the PVEs. What is the result? Authors should justify Lasso selected SNPs are optimal choice for XGBoost.

- If you select SNPs based XGBoost feature importance, and then use lasso for classification, what would be the result?

Reviewer #3 (Remarks to the Author):

In this paper, the authors developed a method utilizing machine learning algorithms to account for non-linear and interaction effects between SNPs in polygenic risk scores construction, which is a welcoming development. The overall presentation of the paper are clear but there are a few points that might need clarifying.

1. The authors have shown that when compared with the C+T method, their XGBoost method can often result in a ~20% relative increase, which is impressive. However, C+T is one of the simplest PRS calculation method, many recent methods, such as lassosum, LDpred2, SBayesR, PRS-CS e.t.c. have been developed, each consistent out-perform the C+T method. Therefore, it is difficult to know if XGBoost performance gain is due to considering the non-additive SNP effects, or simply because C+T method is not performing well. One might need to compare with more advance method(s) in order to know for sure.
2. In some scenario, it is noted that XGBoost under-perform when compared with C+T (e.g. height). Would this be due to pre-filtering of SNPs (the author seems to pre-filter SNPs with p-value less than $1e-4$ for XGBoost)? If we only consider p-value threshold less than $1e-4$ for the C+T method, will XGBoost consistently out-perform C+T method?
3. In similar vein, some existing methods such as lassosum and LDpred2 will return SNP weighting after taken into consideration of LD, and can consider the whole genome at once. Would it be possible to perform, say, lassosum as a first step, obtain SNPs with non-zero weighting, and then use the resulting SNP weighting as input to XGBoost? Or would it be possible to run the LASSO penalized regression on all SNPs without pre-filtering, then only include SNPs with non-zero coefficient for XGBoost?
4. Given the high computational burden of XGBoost, is it possible that highly polygenic traits will tends to have lower performance due to the pre-filtering of SNPs?
5. Is the performance reported independent of the covariate? E.g. the variance explained by the PRS alone, or the variance explained by the full model (covariate and other features included?)
6. Resolution of plots are too low.
7. In the Black population, it is observed that sometimes the XGBoost with PRS underperform compared to C+T PRS, are there any explanation for that? Considering the C+T PRS itself is a feature of the model, one would expect the best performing parameter of the XGBoost with PRS model should at least equal to the C+T PRS model.

Item-by-item responses to reviewer comments

Reviewer #1:

Elgart et al. present an interesting study on polygenic risk scores where they find that gradient boosted trees capture non-linear effects and epistatic interactions. I personally believe that the research question raised is important as it can improve the accuracy of polygenic risk scores and their value in studying genetics and in clinical settings. I also found the paper to be well written and I appreciate the effort made to include multi-ethnic data. However, I would have liked to see a greater comparison of methods, as I believe the benefit seen by gradient boosted trees could be partially due to the few number of variants included in the PRSs, and how they are generated. I therefore have a number of comments that I would like to see addressed in a revised version.

Response: Thank you for taking the time to carefully review this manuscript.

1. Clumping and thresholding (C+T), as implemented in PRSice is as the authors note still a common strategy. However recently multiple methods have been proposed that seem to consistently outperform C+T in benchmark studies (Ni et al., Biol PSych 2021, <https://doi.org/10.1016/j.biopsych.2021.04.018>; Kulm et al., medRxiv 2021, <https://doi.org/10.1101/2020.04.06.20055574>; Pain et al., PLoS Genet 2021, <https://doi.org/10.1371/journal.pgen.1009021>; Zhou and Zhao, PLoS Genet 2021, <https://doi.org/10.1371/journal.pgen.1009697>). I would like to see comparisons with methods like PRS-CS, LDpred2, SBayesR, SDPR, and lassosum (at least one such method). I am curious to see whether the improvement observed here is also observed when using these instead of C+T.

Response: Thank you for this suggestion. We agree with this comment. We have implemented LDpred2 for our cohort and compared the results to the C+T PRS in addition to the XGBoost

models. We describe the methodology in the Methods at page 8-9, lines 16-25. We discuss the results of this analysis in the Results at page 12 lines 14-21; page 13 lines 18-24; and in the new Figure 2. Briefly, LDpred2-based PRS outperformed PRSice-based PRS in 5 out of the 9 phenotypes. XGBoost-based models that include PRS and individual genetic variants are still better than standard linear models with PRS.

2. The authors are absolutely correct that C+T will generally miss many haplotypes and epistatic effects. However, if we believe some of these haplotypes and epistatic effects are captured by neighbouring variation, it is possible that including more variants in the polygenic score can improve the prediction. I am worried that this might in particular be a problem for C+T due to the LD-clumping step, which seems to be chosen to be very strict ($r^2=0.1$). This results in very few variants being actually used when, e.g., predicting height. Hence, I recommend trying different LD clumping parameters, e.g. $r^2=0.2$ and a larger LD window.

Response: Thank you for pointing out this important note. We now compare more potential PRS, in the new Figure 2 (more clumping parameters for PRSice, and LDpred), and choose the best performing (on the test set) PRS out of these. We assessed C+T PRS with r^2 of 0.1, 0.2, and 0.3; and LD window of size 250kb and 500kb. We also assessed LDpred2 using the Auto and Infinitesimal (Inf) methods on both clumped and the top 1 million SNPs. We then select the best-performing PRS for purposes of comparison and training the XGBoost model. These results are found in detail in Figure S2 as well as incorporated to the new Figure 2. Indeed, performance of all models (evaluated on the test data) have improved after this change. These changes are discussed in the Results, page 12 line 22 through page 13 line 5.

2.b. Maybe the strict thresholding applied might lead to a more robust PRS with respect to predicting into multi-ethnic data. That is something one can actually try out on real data, i.e. are the new scores more or less robust (wrt genetic ancestry) when training using individuals of one genetic ancestry and predicting into a sample with a different genetic ancestry.

Response: This is a great point. Your comment could be interpreted either regarding PRS specifically, or regarding our proposed gradient boosted trees model. Focusing on PRS first, our model does not readily allow for this comparison, because the training dataset is highly affected by the GWAS used to estimate genetic associations, and we did not consider multiple possible GWAS with different training populations. With respect to our gradient boosted trees model, we think that our Figure 5 (previously Figure 4) addresses the issue of prediction using multi-ethnic versus race/ethnic-specific data. Specifically, for each phenotype, we trained our models separately on the multi-ethnic cohort, Black participants, Hispanic/Latino participants, and White participants; and we then predict into the test set of each race/ethnicity group. For example, we can see that for triglycerides, when trained on the full multi-ethnic cohort, the test PVE was 9.3% among White participants, 6.8% among Black participants, and 14.8% among Hispanic/Latino participants; when we trained the model on only White participants, however, the test PVE is 9.7% among White participants, 5.1% among Black participants, and 9.4% among Hispanic/Latino participants.

3. Regarding citation 16, they report in my opinion a suspiciously high AUC for CAD PRS. Alternative citations pointing out the value of capturing non-linear effects could include Sigurdsson et al., (bioRxiv 2021, <https://doi.org/10.1101/2021.06.11.447883>), but they find that NNs can improve prediction over lasso for immune-related diseases, but importantly, not all diseases.

Response: We have updated citation 16 and the text describing it, on page 4, line 12-13: “Other studies have found that deep neural networks outperform linear models for a wide range of diseases, but not all.”, referencing Sigurdsson et al.

4. In line 103-104 you say “we hypothesize that a large and ancestry diverse cohort would improve genetic prediction across populations”. Although I intuitively agree with this, I believe the proof is in the pudding. More specifically, I would like to see whether this is actually the case for the analysis proposed here, or maybe citations that do examine this in details, e.g. by trying different approaches.

Response: Thank you for this comment. We believe this to be an important analysis in our work. The results are found in Figure 5 (previously Figure 4). We discuss the results on page 16, lines 5-13, and page 18 line 18 through page 19 line 16. Our analysis confirms that it is useful to have data from diverse populations to generate prediction models. However, contrary to our expectation, models that are trained on race/ethnically-matched sets of individuals to those in the test set usually performed equally well, and sometimes better, compared to models trained on a larger dataset that included more individuals, from additional race/ethnicities. Therefore, we cannot rule out that the improvement that we see when using multi-ethnic populations in the training dataset is due to increased sample size, rather than due to diversity.

5. You restrict the analysis to 5 complex outcomes. Why not more, so that the statement could become more general. In particular, I would be interested in immune-related diseases. I would also like to see whether your results hold in UKB data, if possible. The UKB would also enable you to have “large” training data.

Response: Thank you for this suggestion. We agree that additional phenotypes would strengthen this work. Therefore, we have added an additional 4 phenotypes to the analysis: body mass index (BMI), HDL Cholesterol, LDL Cholesterol, and Diastolic Blood Pressure. The results for these phenotypes are discussed throughout the manuscript and visualized in Figures 2, 3, 4, and 5. With regard to the UKB, unfortunately we did not have the resources (budget to purchase data access and required computing storage) to add an analysis using UK Biobank. We will certainly aim to expand our work to UKB in the future.

6. It is unclear to me how much overlap there was between the variants available in the GWAS sum stats and the 705M variants in the TOPMed data. Also, what LD reference did you use for clumping in PRSice? I am worried that using multi-ethnic GWAS sum stats can create problems when adjusting for LD in the LD-clumping step. The LD reference should ideally reflect the LD in the GWAS sample, which in practice means that the ancestry composition of the GWAS sample should be similar to the one in the LD reference sample. I would appreciate details on this as it will impact the accuracies of the polygenic scores. (It would of course be best if LD references

were always released together with the GWAS sum stats, but that's currently not the case.)

Response: The TOPMed data has 705M variants before filtering and QC. We extracted allele counts of variants that passed QC from GDS files using the SeqArray package version 1.28.1 and then further processed using R and Python scripts. We now provide this information in the Supplemental Table S14, which contains the number of variants that overlap between the TOPMed data and the GWAS data, and in page 6 lines 17-19: "After QC and filtering variants with $MAF < 0.01$ (with MAF being computed based on the multi-ethnic TOPMed dataset), we had 12,482,699 variants in the TOPMed data." And page 7 line 22 through page 8 line 2: "For about half of the phenotypes, 90% or more of the variants in the GWAS were found in the TOPMed data. For the other half, 60-70% were found (Table S14)."

We used the entire multi-ethnic TOPMed data to perform clumping with PRSice and to compute SNP weights using LDpred2. Not written in the paper, in a different ongoing work, we saw PRSice is not sensitive to the choice of reference panel while LDpred, as expected, tend to perform better (though not always) when applied on a reference panel with similar genetic ancestry to the discovery GWAS.

6.b Also, what additional filters did you apply, e.g. MAFs, etc.

Response: Thank you for bringing this up. We require $MAF \geq 0.01$, with MAF being computed based on the multi-ethnic TOPMed dataset. Otherwise, we used TOPMed Quality filters, described in the sequencing methods in this link:

<https://www.nhlbiwgs.org/topmed-whole-genome-sequencing-methods-freeze-8>

We have clarified these filters in the Methods, page 6 lines 15-19: "We extracted allele counts of variants that passed QC from GDS files using the SeqArray package version 1.28.1 and then further processed using R and Python scripts. After QC and filtering variants with $MAF < 0.01$ (with MAF being computed based on the multi-ethnic TOPMed dataset), we had 12,482,699 variants in the TOPMed data. For all variants, we set the effect allele to be the minor allele."

6.c How did you deal with variants with ambiguous alleles, C/G and A/T, when matching variants between the GWAS sum stats and the TOPMed data?

Response: We used the standard PRSice software to remove ambiguous alleles. We have clarified this in the manuscript at page 8 lines 6-7: “SNPs with ambiguous alleles were removed using the PRSice software.”

7. PRSice by default tries extremely many thresholds, which I have found to lead to overfitting problems in some instances. To address this, one can either reduce the number of thresholds tested, e.g. try only 10-20 thresholds, or one can use some methods that do not require validation data, for fitting hyperparameters, e.g. SBayesR, PRS-CS-auto, LDpred2-auto, etc.

Response: We used 10 thresholds for each combination of clumping parameters with PRSice and selected the threshold and clumping parameters that minimized the mean square error in the training dataset. The reported results are from the test dataset. We report results from the test dataset (not the training dataset) to ensure that our results are not overfitting. Finally, we now added LDpred2-auto and -inf. We clarified this point at page 8 lines 10-14: “We considered p-value thresholds of 0.5 through $1e-10$. For each adjusted phenotype and each PRS defined by clumping region, clumping R^2 value, and p-value threshold, we fit a linear model including covariates, the PRS, and genetic PCs to account for population structure ³³. We selected the PRS where the PRS model minimized the mean squared error in the training dataset.”

8. Equation in line 241 is sometimes referred to as adjusted R^2 .

Response: We added a note to clarify this in the text at page 11 lines 17-18: “We quantify model performance as the variance explained (sometimes referred to as the adjusted R^2).”

Reviewer #2 (Remarks to the Author):

The main claim of the study is that combining linear and non-linear machine learning models better works for phenotype predictions based on genetic data:

- How SNPs are encoded before inputting to the models? Have you tested changing the encoding scheme and look at results again? You may get different sets of potentials SNPs to input to the models.

Response: Thank you for raising this point. We coded SNPs as counts of the minor alleles. It is true that we will likely get a different set of SNPs if we also tried recessive and dominant coding, however, the standard in the field is to use additive coding which tends to be the most powerful (statistically). The XGBoost model later on can automatically account for potential non-additivity, and in more sophisticated ways compared to just additivity and dominance.

- Can authors justify why adding a single feature as PRS to a stochastic model as XGBoost considerably increases the PVE values?

Can you demonstrate the distribution of prediction scores of the classes before and after adding PRS to the XGBoost model?

Response: Thank you for raising this important question. We agree that it is an important discussion point, and it is discussed in the Discussion at page 17 lines 2-8: "In all cases, the XGBoost algorithm alone (without including the PRS score) out-performed the linear LASSO model that used exactly the same SNPs. In half the phenotypes, however, the linear PRS performed better, likely because it could account for more weakly-associated SNPs. Combining the ML model with the PRS (as a feature) achieved high prediction performance by both accounting for the large numbers of weakly associated SNPs (linearly through PRS), in addition to some of the non-linearities and interactions (through XGBoost)."

As well as in our Limitations at page 20 lines 11-13: "Finally, due to the high complexity of the XGBoost model and pre-filtering of SNPs, highly polygenic traits (such as Height) may suffer in performance when compared to less complex phenotypes."

We believe that combining the ML model with a PRS (as a feature) achieved high prediction performance because the PRS is highly associated with the outcome by accounting for the large numbers of weakly associated SNPs, which could not be incorporated as individual features in

the XGBoost due to overfitting. The XGBoost further accounts for some of the non-linearities and interactions through individual SNPs.

Because these are continuous phenotypes, there are no classes to examine. Instead, we examined the held-out test set Percentage of Variance Explained, which is the comparable metric for a continuous phenotype.

- One of the draw backs of the study is that the authors still consider known SNPs that have been identified by GWAS, and as they have also mentioned GWAS SNPs have been computed one-by-one without considering interaction between them. From the material, authors have over 7m SNPs. How do you consider the interaction between non-GWAS selected SNPs and generally unknown SNPs?

Response: Thank you for this insightful comment. We agree that this is a limitation of this study, and an important topic of future work. We have expanded on this in the Discussion, at page 19 lines 23-24: "Similarly, we have not considered the interactions between non-GWAS selected SNPs, which may lead to some important variants with interaction effects being excluded."

- What is the loss function for XGBoost and Lasso? Is the task classification or regression? If classification, why you considered MSE loss, that is for regression tasks?

Response: This is a regression task with continuous phenotypes. The loss function for the XGBoost and LASSO models is the Mean Square Error (MSE). We have clarified this point in the Methods at page 10 line 17: "This was a regression task." and page 10 line 8-11: "Based on this 3-fold cross validation, we selected the number of trees θ_α that minimized the mean squared prediction error (MSE), resulting in a set of parameters (α, θ_α) . We selected the optimal (α, θ_α) pair that minimized the MSE of the 3-fold cross validation step across all values of α ."

- Can you repeat an experiment with randomly selecting SNPs same size as the Lasso selected SNPs and then add PRS feature to it? Please repeat this randomized experiment for 100-1000 times and average the PVEs. What is the result? Authors should justify Lasso selected SNPs are optimal choice for XGBoost.

Response: Thank you for this interesting suggestion. However, we believe that our approach to SNP selection using LASSO is superior to random selection. We ensure that the LASSO selected SNPs are the optimal choice for XGBoost by utilizing a joint training methodology of the LASSO / XGBoost ensemble model that utilizes the loss function for XGBoost when selecting the optimal L1 regularization hyperparameter for LASSO. We do agree with the reviewers that other strategies to select SNPs may be better, particularly strategies that may prioritize SNPs that have non-linear and interaction effects. In fact, in the Discussion, page 20 lines 1-4, we wrote “Third, much of our ensemble algorithm relies on feature selection. This may be overly restrictive and does not allow for variants with very small effect sizes to be included (as noted in the results for Height). It is also possible that SNPs selected through LASSO may not be prioritized based on non-linear or interaction effects, even though we model them using the non-linear XGBoost.” We did not employ such strategies in this work because they are a topic of future methodological development (i.e., no existing approaches exist yet).

- If you select SNPs based XGBoost feature importance, and then use lasso for classification, what would be the result?

Response: We did initially investigate training XGBoost on the full set of available SNPs. We found that the XGBoost model was prone to overfitting with such high dimensionality, and therefore we created our custom joint training methodology that optimizes the SNPs selected by LASSO with respect to the XGBoost loss function. We found that the joint training of the ensemble model significantly outperformed XGBoost alone without any ensemble penalization methodology.

Reviewer #3 (Remarks to the Author):

In this paper, the authors developed a method utilizing machine learning algorithms to account for non-linear and interaction effects between SNPs in polygenic risk scores construction, which

is a welcoming development. The overall presentation of the paper are clear but there are a few points that might need clarifying.

Response: Thank you for carefully reviewing our manuscript and providing these helpful comments.

1. The authors have shown that when compared with the C+T method, their XGBoost method can often result in a ~20% relative increase, which is impressive. However, C+T is one of the simplest PRS calculation method, many recent methods, such as lassosum, LDpred2, SBayesR, PRS-CS e.t.c. have been developed, each consistent out-perform the C+T method. Therefore, it is difficult to know if XGBoost performance gain is due to considering the non-additive SNP effects, or simply because C+T method is not performing well. One might need to compare with more advance method(s) in order to know for sure.

Response: Thank you for this suggestion. We agree with this comment. We have implemented LDpred2 for our dataset and compared the results to the C+T PRS as well as XGBoost. We also tried out more clumping parameters for PRSice and various hyperparameters for LDpred2, and chose the best performing (on the training set) PRS out of these. We assessed C+T PRS with \$r^2\$ of 0.1, 0.2, and 0.3; and LD window of size 250kb and 500kb. We also assessed LDpred2 using the Auto and Infinitesimal (Inf) methods on both clumped and the top 1 million SNPs. We then select the best performing PRS for purposes of comparison and training the XGBoost model. We describe the methodology in the Methods at page page line 5 through page 9 line 5. We discuss the results of this analysis in Results at page 12 line 14 through page 13 line 8; and in the new Figure 2, in detail in Figure S2, and Table S10. To summarize, using these additional PRS increased the performance of all models on the test set, and XGBoost + PRS still improves over the linear PRS model and the “XGBoost alone” model.

2. In some scenario, it is noted that XGBoost under-perform when compared with C+T (e.g. height). Would this be due to pre-filtering of SNPs (the author seems to pre-filter SNPs with p-value less than $1e-4$ for XGBoost)? If we only consider p-value threshold less than $1e-4$ for the C+T method, will XGBoost consistently out-perform C+T method?

Response: We believe that this phenomenon occurs because the C+T PRS is able to account for a large number of weakly associated SNPs (linearly), while the XGBoost model tended to overfit for more complicated phenotypes with many weakly associated SNPs contributing to the overall effect. We believe that by combining the ML model with the PRS (as a feature), we were able to account for this effect in complicated phenotypes while additionally accounting for some of the non-linearities and interactions (through XGBoost).

We also compared the C+T PRS with p-values less than $1e-4$ directly to XGBoost, so as to use the same set of candidate SNPs. We have reported results for this analysis in Supplemental Table S15 and discussed in the manuscript at page 14 lines 1-7: “The results were qualitatively similar to those in the primary analysis: on most cases, the XGBoost alone model outperformed the linear PRS model, but not for BMI, height, and HDL cholesterol. The latter two had a larger number of SNPs, likely leading to overfitting (Table 3).”

3. In similar vein, some existing methods such as lassosum and LDpred2 will return SNP weighting after taken into consideration of LD, and can consider the whole genome at once. Would it be possible to perform, say, lassosum as a first step, obtain SNPs with non-zero weighting, and then use the resulting SNP weighting as input to XGBoost? Or would it be possible to run the LASSO penalized regression on all SNPs without pre-filtering, then only include SNPs with non-zero coefficient for XGBoost?

Response: if we understand correctly, the reviewer is asking about a different way to select and input SNPs into the XGBoost model while considering LD between SNPs. We would like to clarify the following, which we believe does address the reviewer comments. First, we did perform a sensitivity analysis, reported in Table S4 in the Supplementary Information, where we clumped SNPs before applying LASSO, addressing the LD effect (the SNPs are not clumped in the main results). Notably, the results were very similar. Second, SNP weighting should not impact the XGBoost results: multiplying any feature by a constant does not affect a tree-based model (while, in contrast, it would have an effect on a PRS). While we could try additional methods to select SNPs into the XGBoost models, e.g. filtering by weight, we do not believe that it would improve results as this would not encourage, more so than LASSO, the selection of

SNPs that are more likely to have non-linear or interaction effects. Therefore, we respectfully chose to continue with the LASSO approach as a principally valid approach for variable selection. Finally, while not reported, we did attempt performing LASSO penalized regression on all SNPs. However, this was not computationally feasible, likely because it requires more sophisticated computational resources than we have access to.

4. Given the high computational burden of XGBoost, is it possible that highly polygenic traits will tend to have lower performance due to the pre-filtering of SNPs?

Response: Thank you for raising this important point. We do believe that this may be a limitation of this work, and it is exemplified in the results that we see for height, a highly polygenic trait, for which XGBoost alone performs worse when compared to the C+T PRS. We have clarified this limitation in the Discussion, page 20 lines 11-13: “Finally, due to the high complexity of the XGBoost model and pre-filtering of SNPs, highly polygenic traits (such as Height) may suffer in performance when compared to less complex phenotypes.”

5. Is the performance reported independent of the covariate? E.g. the variance explained by the PRS alone, or the variance explained by the full model (covariate and other features included?)

Response: Correct. We compared to a null model that includes covariates, so the performance is the addition of variance explained after adjusting for covariates. This is described on page 7 lines 12-15: “Then, each phenotype was regressed on age, sex, study, and race/ethnicity. The residuals were extracted and rank-normalized. Subsequent analyses used these rank-normalized residuals as the outcomes, and we refer to them henceforth as “adjusted phenotypes”.”

6. Resolution of plots are too low.

Response: Thank you for pointing this out. We have re-uploaded the Figures with higher resolution.

7. In the Black population, it is observed that sometimes the XGBoost with PRS underperform compared to C+T PRS, are there any explanation for that? Considering the C+T PRS itself is a feature of the model, one would expect the best performing parameter of the XGBoost with PRS model should at least equal to the C+T PRS model.

Response: Thank you for raising this important observation. We also find this surprising; however, this phenomenon is for predictions of the triglycerides and body mass index phenotypes only. There are a few possible explanations. We have expanded on this in the Discussion, at page 17 line 18 through page 18 line 2: “Surprisingly, among Black participants, the XGBoost with PRS model underperforms compared to the linear PRS for two phenotypes: triglycerides (PVE of 7.1% for linear PRS and 6.8% for XGBoost with PRS among Black participants) and body mass index (PVE of 1.7% for linear PRS and 1.4% for XGBoost with PRS among Black participants). There are a few possible explanations for this phenomenon. It’s possible that there is more genetic diversity in the Black population compared to the other race/ethnic groups. It’s also possible that the smaller sample size of the Black population led to more severe overfitting and less generalizability of results.”

Reviewers' comments:

Reviewer #1 (Remarks to the Author):

I would like to thank the authors for addressing all of my comments. The only substantial comment that I have left is a question about how LDpred2 was actually used. It seems to me that the authors first applied some LD-clumping before running LDpred2. However, if true, this will in general induce a winner's curse bias for any Bayesian PRS method (SBayesR, PRS-CS, LDpred2, etc.), unless accounted for. See e.g. Shi et al., PLoS Gen 2016 or Privé et al., AJHG 2022, for how to account for winner's curse. Alternatively you can just do (stupid) LD-pruning (ignoring variant p-values) instead of LD-clumping. I suspect this could have a large impact on the LDpred2 results presented here.

Reviewer #2 (Remarks to the Author):

Thank the authors for addressing the comments. I recommend a minor revision before publications by addressing these:

'However, we believe that our approach to SNP selection using LASSO is superior to random selection.'

I would have liked to see such experiment as baseline experiments. Because of high dimensionality of SNPs, LASSO might still select SNPs that are not correlated with the outcome and rather there might be selected randomly. I think that the selected SNPs' coefficients in the LASSO algorithm are very small and towards to be values near zero.

'- If you select SNPs based XGBoost feature importance, and then use lasso for classification, what would be the result?'

Studies such as <https://www.nature.com/articles/s41598-018-31573-5> and <https://www.nature.com/articles/s41598-020-66907-9> have also taken into account non-linear interaction between SNPs. Most importantly, the trained model in their studies are not based on pre-selected GWAS SNPs. I strongly suggest discuss such studies in the manuscript.

Reviewer #3 (Remarks to the Author):

In this update the authors have provided new analyses to show that XGBoost are a better PRS analyses by accounting for non-linear and interaction effects. However, the authors failed to answer some of my concerns and there are some major flaws in the new analyses, specifically related to the usage of LDpred2. Without addressing these issues, I cannot recommend this paper for publication.

1. Regarding my previous comment on lassosum: most of the modern PRS algorithms, e.g. lassosum, LDpred2, PRS-CS, SBayesR, that can adjust for SNP effect size while considering the LD structure across the whole genome. These methods do more than just multiplying a constant to the effect sizes and are expected to provide much higher performance than traditional C+T method. For example, in UK biobank, C+T will generate a R2 of ~18% for height, whereas LDpred will generate a R2 of ~21% and lassosum will generate a R2 of ~24%. As such, my previous question was, if instead of just performing LASSO on a set of pre-selected SNPs, how would XGBoost be affected if the input effect sizes has been adjusted using dedicated software?

2. In similar vein, while the authors correctly stated that methods such as LDpred2 and PRS-CS can

account for LD reference panels when estimating the joint SNP effects, which is an advantage over the traditional C+T method, I am confused as to why clumping / selection were performed prior to their LDpred2 analyses, which is not the standard procedure and counter intuitive. This can reduce the performance of LDpred2 and might not provide a fair comparison.

3. There are three models for LDpred2: auto, grid and inf. It is usually recommended to test all three models and select one that performs best. Can the authors also include the grid model?

4. The phenotype might need more work. Is race/ ethnicity categorical information, or were those represented by the 5 PCs? From p7 line 10, the author stated that "For each of the covariate adjusted phenotypes of interest", and then from line 12-13 "each phenotype was regressed on age, sex, study, and race/ethnicity". How can the author filter samples based on the covariate-adjusted phenotype, before they perform the covariate adjustment?

5. If the residualized phenotype were used for the downstream analyses, why do the authors include the covariates into the linear model? If covariates were included in the linear model, how do the authors obtain the PRS specific performance?

6. It would be better if the authors stress that the performance increase is a "relative increase". Current phrasing sounds as if the improvements are on absolute scale: "... XGBoost with PRS improved the PVE by 22% ..."

7. If XGBoost model is prone to overfitting with high dimension data, how can a user identify the "sweet-spot" of amount of information vs risk of overfitting when selecting variants for XGBoost? From my understanding, XGBoost model requires 2 stages of selection: 1) Select top variants with p-value less than 1×10^{-4} , then perform lasso; 2) Run the XGBoost prediction model. For more polygenic traits / more powerful GWAS, should one use a more stringent threshold?

Response to review of COMMSBIO-21-1854A: “Polygenic Risk Prediction using Gradient Boosted Trees Captures Non-Linear Genetic Effects and Allele Interactions in Complex Phenotypes”

We thank the reviewers for reviewing our manuscript and for their suggestions, and to the editor for giving us the opportunity to address these additional comments. Below we provide an item-by-item response to the reviewers’ comments. We have updated the manuscript, figures, and tables according to the additional analyses completed; we also removed page 17 lines 18-23 from the previous submission as it is no longer relevant due to our changes. We use blue font for our responses and explanations.

Reviewer #1 (Remarks to the Author):

I would like to thank the authors for addressing all of my comments. The only substantial comment that I have left is a question about how LDpred2 was actually used. It seems to me that the authors first applied some LD-clumping before running LDpred2. However, if true, this will in general induce a winner's curse bias for any Bayesian PRS method (SBayesR, PRS-CS, LDpred2, etc.), unless accounted for. See e.g. Shi et al., PLoS Gen 2016 or Privé et al., AJHG 2022, for how to account for winner's curse. Alternatively you can just do (stupid) LD-pruning (ignoring variant p-values) instead of LD-clumping. I suspect this could have a large impact on the LDpred2 results presented here.

Response:

Thank you for raising this important point. We indeed corrected for the winner’s curse prior to applying LDpred2.

Regarding LD-pruning and otherwise filtering of SNPs to reduce the number of SNPs that LDpred will use: LDpred2 can use about 1-2M SNPs. We tried to approaches to limit the number of SNPs. (a) We pruned (as you suggested, meaning, we did not clump) SNPs using pruning parameters $R^2=0.1$ and distance=500Kb; and (b) We used the top 1 million SNPs with respect to their association p-value in the summary statistics (and that were also available in the TOPMed dataset) to train the model. The decision to move forward with these approaches was based on the proposed approaches in a preprint by the authors of LDpred2 (<https://www.biorxiv.org/content/10.1101/2021.03.29.437510v2>), evaluating computational limitations, and correspondence with the author of LDpred 2 on GitHub (<https://github.com/privéfl/bigsnpr/issues/269>). Note that we could not implement another proposed approach by LDpred2 author (considered in the same preprint), to use HapMap SNPs, because high proportion of these SNPs had (low) levels of missing values in our TOPMed dataset. Finally, we applied the winner’s curse correction for both methodologies.

Quote: "... we used the bigsnpr R package to correct the estimated effect sizes for the "winner's curse" prior to applying LDPred2."

Page, Line: page 9, lines 3-5

Reviewer #2 (Remarks to the Author):

Thank the authors for addressing the comments. I recommend a minor revision before publications by addressing these:

"However, we believe that our approach to SNP selection using LASSO is superior to random selection."

I would have liked to see such experiment as baseline experiments. Because of high dimensionality of SNPs, LASSO might still select SNPs that are not correlated with the outcome and rather there might be selected randomly. I think that the selected SNPs' coefficients in the LASSO algorithm are very small and towards to be values near zero.

Response:

Thank you for this suggestion to establish a baseline. We have performed this experiment for four phenotypes (total cholesterol, triglycerides, LDL cholesterol, and HDL cholesterol), by 1) randomly selecting SNPs in the same size as the LASSO selected SNPs for those phenotypes, 2) running the XGBoost model with and without PRS, 3) repeating 100 times, and 4) averaging the result.

The results are found in Table S16 and discussed in the manuscript.

Quote (Methods section):

"... we selected SNPs at random as a baseline for four phenotypes (total cholesterol, triglycerides, LDL cholesterol, and HDL cholesterol). We used the same number of SNPs as the number selected by LASSO in the respective XGBoost Alone and XGBoost with PRS models. We have performed this experiment for four phenotypes (total cholesterol, triglycerides, LDL cholesterol, and HDL cholesterol), by 1) randomly selecting SNPs in the same size as the LASSO selected SNPs for those phenotypes, 2) running the XGBoost model with and without PRS, 3) repeating 100 times, and 4) averaging the result."

Page, Line: page 11 line 19-page 12 line 1

Quote (Results section):

"In Supplementary Table S16, we report results from random SNP selection as a baseline for four phenotypes (total cholesterol, triglycerides, LDL cholesterol, and HDL cholesterol). LASSO selected SNPs are significantly superior to random selection in the XGBoost Alone model, by 20%-175%. For XGBoost with PRS, the increase from LASSO is more attenuated, at only 7%-21% higher than random selection."

Page, Line: Table S16, page 15 lines 12-16

"- If you select SNPs based XGBoost feature importance, and then use lasso for classification, what would be the result?"

Response:

Thank you for this suggestion. We agree that this is a promising idea, as the XGBoost model might select different SNPs than the LASSO. We have attempted to run this. However, due to the high dimensionality of the SNPs and the heavy computational load required by XGBoost, we are unable to perform this experiment due to memory and compute constraints in our cloud infrastructure. This approach would require a significant change to the algorithm's back-end functionality and our cloud computing environment. We believe this is a promising avenue of future research and have added it to the Discussion.

Quote: "It is also possible that SNPs selected through LASSO may not be prioritized based on non-linear or interaction effects, even though we model them using the non-linear XGBoost. A promising area of future research could be using XGBoost on the full set of candidate SNPs to perform feature selection, and then use LASSO (or another algorithm) for prediction of classification, while potentially including interaction terms and other SNP models (dominant, recessive) as features."

Page, Line: page 21 lines 11-16

Studies such as <https://www.nature.com/articles/s41598-018-31573-5> and <https://www.nature.com/articles/s41598-020-66907-9> have also taken into account non-linear interaction between SNPs. Most importantly, the trained model in their studies are not based on pre-selected GWAS SNPs. I strongly suggest discuss such studies in the manuscript.

Response:

Thank you for this suggestion. The first article you suggest, "Machine learning identifies interacting genetic variants contributing to breast cancer risk: A case study in Finnish cases and controls", was already discussed in our manuscript (citation 17), but we have expanded the reference in both the Introduction and the Discussion, and additionally added in the second article.

Quote:

Introduction: "Gradient boosted trees have been used to predict breast cancer risk by first identifying nonlinear SNP-SNP interactions using XGBoost or networks and then using support vector machines for discrimination, which resulted in increased mean average precision when compared to generalized linear models."

Page, Line: page 4 lines 11-15

Discussion: “A limitation of our current computational infrastructure is the inability to run XGBoost on many hundreds of thousands of SNPs, which, if ameliorated, would allow us to use XGBoost for feature selection, as some other studies have done ^{17,18}.”

Page, Line: page 21 lines 16-19

Reviewer #3 (Remarks to the Author):

In this update the authors have provided new analyses to show that XGBoost are a better PRS analyses by accounting for non-linear and interaction effects. However, the authors failed to answer some of my concerns and there are some major flaws in the new analyses, specifically related to the usage of LDpred2. Without addressing these issues, I cannot recommend this paper for publication.

1. Regarding my previous comment on lassosum: most of the modern PRS algorithms, e.g. lassosum, LDpred2, PRS-CS, SBayesR, that can adjust for SNP effect size while considering the LD structure across the whole genome. These methods do more than just multiplying a constant to the effect sizes and are expected to provide much higher performance than traditional C+T method. For example, in UK biobank, C+T will generate a R2 of ~18% for height, whereas LDpred will generate a R2 of ~21% and lassosum will generate a R2 of ~24%. As such, my previous question was, if instead of just performing LASSO on a set of pre-selected SNPs, how would XGBoost be affected if the input effect sizes has been adjusted using dedicated software?

Response:

The original remark from the Reviewer was: “Would it be possible to perform, say, lassosum as a first step, obtain SNPs with non-zero weighting, and then use the resulting SNP weighting as input to XGBoost?”

Thank you for this suggestion. That is a very interesting idea to help account for the SNP selection problem in a novel way. We have implemented your idea in the following manner:

First, we have implemented lassosum and added it to Figure 2, Figure S2, Table 3, Table S8, and Table S10. Lassosum was superior to PRSice and LDpred2 for two phenotypes: triglycerides and BMI.

For these two phenotypes (triglycerides and BMI), we then re-implemented the algorithm using lassosum as the best PRS and updated Figure 3, Figure 4, Figure 5, Table S11, Table S12, and Table S13.

For the phenotypes, we found that lassosum chose a large number of SNPs, usually in the tens or hundreds of thousands, up to 249,700 total SNPs for systolic blood pressure. For triglycerides the lassosum selected over 8,000 SNPs and for BMI the lassosum selected 51,000. We are unable to run the XGBoost model on tens or hundreds of thousands of SNPs due to the heavy computational load with constraints in our cluster computing infrastructure.

We were, however, able to perform this experiment for LDL Cholesterol, which only selected 2,056 SNPs through lassosum. The results are in Table S17 and discussed in the manuscript. We found that using the lassosum selected SNPs resulted in a test EVR of 9.0% compared to 13.3% when using the LASSO model.

Quotes:

Lassosum PRS - Methods: “Finally, we calculated PRS via penalized regression on summary statistics, lassosum³⁸. We used the lassosum2 implementation in R packages bigsnpr v1.9.5 and bigstatsr v1.5.6³⁹, using the default hyperparameters as described in detail in Prive et al.⁴⁰”

Page, Line: page 9 lines 6-8

Lassosum PRS – Results: “Lassosum was superior to LDpred2 and PRSice for two phenotypes: triglycerides (12% higher than PRSice PRS) and body mass index (15% higher than PRSice PRS).

Page, Line: page 13 lines 19-21, Figure 2, Figure S2, Table 3, Table S8, and Table S10

Results for XGBoost Models with lassosum PRS (where relevant): (found in Figures/Tables and the Results are updated throughout the text)

Page, Line: Figure 3, Figure 4, Figure 5, Table S11, Table S12, and Table S13

Lassosum selected SNPs - Methods: “In the second experiment, for LDL cholesterol we used the SNPs with non-zero weighting in the lassosum PRS as the selected SNPs in the XGBoost model.”

Page, Line: page 12 lines 1-4

Lassosum selected SNPs – Results: “In Supplementary Table S17, we report results from using SNPs selected into lassosum PRS for LDL cholesterol. The test EVR for the XGBoost model with the lassosum selected SNPs was 9.0% compared to 13.3% when using the LASSO SNP selection model.”

Page, Line: page 15 lines 16-19, Table S17

2. In similar vein, while the authors correctly stated that methods such as LDpred2 and PRS-CS can account for LD reference panels when estimating the joint SNP effects, which is an advantage over the traditional C+T method, I am confused as to why clumping / selection were performed prior to their LDpred2 analyses, which is not the standard procedure and counter intuitive. This can reduce the performance of LDpred2 and might not provide a fair comparison.

Response:

We thank the reviewer for this comment. LDpred2 can compute joint weights for up to only ~1 million SNPs. We considered a few approaches for SNP filtering, including via an open GitHub discussion with the author of the LDpred2 algorithm (<https://github.com/privefl/bigsnpr/issues/269>), who also is studying this problem and reports the selection of a lower number of SNPs (<https://www.biorxiv.org/content/10.1101/2021.03.29.437510v2>). We attempted to two approaches for SNP selection: (a) we pruned SNPs using pruning parameters $R^2=0.1$ and distance=500Kb; and (b) the top 1 million SNPs with respect to their association p-value in the summary statistics (and that overlapped with the TOPMed dataset) to train the model. We did not implement another approach used by the LDpred2 author, which is to use HapMap SNPs and add an additional set of SNPs. This is because a large proportion of HapMap SNPs had (low) level of missingness in TOPMed, and the package does not handle missing values. Finally, we applied the Winner's Curse for both methodologies.

Quote: "Because LDpred2 can compute joint weights for up to ~1 million SNPs, we used two approaches to select SNPs: (a) we pruned SNPs using pruning parameters $R^2=0.1$ and distance=500Kb; and (b) the top 1 million SNPs with respect to their association p-value in the summary statistics (and that overlapped with the TOPMed dataset) to train the model. Following the recommendation provided by LDpred2 manuscript, we used the bigsnpr R package to correct the estimated effect sizes for the "winner's curse" prior to applying LDpred2."

Page, Line: page 8 line 24-page 9 line 5

3. There are three model for LDpred2: auto, grid and inf. It is usually recommended to test all three models and select one that perform best. Can the authors also include the grid model?

Response:

Thank you for pointing this out. We have added in the grid method when running LDpred2. The results can be found in Figure S2. We found that LDpred2-grid was not

better than LDpred2-inf and LDpred2-auto; and for six out of the nine phenotypes the algorithm failed to converge.

Quote:

“LDpred2-auto was superior to LDpred2-inf and LDpred2-grid for seven out of nine phenotypes”

Page, Line: page 14 lines 6-8, Figure S2

4. The phenotype might need more work. Is race/ ethnicity categorical information, or were those represented by the 5 PCs? From p7 line 10, the author stated that “For each of the covariate adjusted phenotypes of interest”, and then from line 12-13 “each phenotype was regressed on age, sex, study, and race/ethnicity”. How can the author filter sample based on the covariate-adjusted phenotype, before they perform the covariate adjustment?

Response:

Thank you for noticing this typo in the description of the phenotype filtering. We have corrected the mistake. The filtering was based on the raw phenotypes.

Separately, regarding your question about adjustment for race/ethnicity, we have adjusted for both race/ethnicity categorical information (White, Black, Hispanic/Latino) and additionally adjusted for genetic ancestry using the 5 PCs.

Quote: “For each of the phenotypes of interest, we excluded outlying individuals defined by phenotypic values above the 99th quantile and values below the 1st quantile for the phenotype, computed over the multi-ethnic dataset.”

Page, Line: page 7 lines 12-14

5. If the residualized phenotype were used for the downstream analyses, why do the authors include the covariates into the linear model? If covariates were included in the linear model, how do the authors obtain the PRS specific performance?

Response:

We residualized the phenotypes to compute the null variance. We then use the covariates again in the linear model in order to obtain accurate inference about the association between the PRS and the outcome, because the PRS wasn't regressed over the covariates. Only if both the exposure and outcome of interest are regressed over the same set of covariates, then we can regress the covariate-adjusted outcome over the covariate-adjusted exposure and obtain the same results exactly as from the regression of the outcome over the exposure while adjusting to covariates. This result is known as the Frisch–Waugh–Lovell theorem. Also see derivation and details in

<https://pubmed.ncbi.nlm.nih.gov/30653739/>, a paper which we cited when referring to the second covariate-adjustment of the covariate-adjusted outcome.

Finally, we obtain the PRS-specific performance by comparing the residual variance from the model with and without the PRS. From the first model without the PRS we compute the variance of the covariate-adjusted phenotype (these are the residuals from the first model). From the second model with the PRS we compute the variance of the covariates-and-PRS-adjusted phenotype. Note that the covariates in the second model should not explain additional variance (meaning, would not lead to inflated variance explained by the PRS) because the phenotypes were already regressed on them.

6. It would be better if the authors stress that the performance increase is a “relative increase”. Current phrasing sounds as if the improvements are on absolute scale: “... XGBoost with PRS improved the PVE by 22% ...”

Response:

Thank you for this comment, we agree that it is better to be clear that this is a relative increase. We have updated the language throughout the manuscript, e.g. “Combining a PRS as a feature in an XGBoost model allowing for non-linear and interaction effects between SNPs **results in a relative increase in the percentage variance explained (PVE)** compared to the standard linear PRS by [...]”

We have updated the language almost everywhere we discuss the relative increase in PVE.

Additionally, we have updated the Methods to describe the method of calculating the relative increase in PVE.

Quote: “We compute relative the PVEs between various models as the relative percentage increase, i.e. $(PVE_2 - PVE_1) / PVE_1$.”

Page, Line: page 13 lines 1-2; and throughout

7. If XGBoost model is prone to overfitting with high dimension data, how can a user identify the “sweet-spot” of amount of information vs risk of overfitting when selecting variants for XGBoost? From my understanding, XGBoost model require 2 stage of selection: 1) Select top variants with p-value less than 1×10^{-4} , then perform lasso; 2) Run the XGBoost prediction model. For more polygenic traits / more powerful GWAS, should one use a more stringent threshold?

Response:

Thank you for this thoughtful comment, it brought up many important issues that we discussed, including, is there a potential relationship between the sample size and the number of features that XGBoost can use without overfitting “too much”? do more polygenic traits tend to have less/more/equal likelihood of SNP-SNP interactions? does

reducing the number of SNPs considered for a highly polygenic traits (while still using the PRS in the XGBoost model) in general still provide some improvement in prediction over a linear PRS model? Another question could be whether other implementation of boosted trees (such as LightGBM which has different tuning parameters) could be more robust to overfitting and what would be their tradeoff with prediction accuracy when they are implemented on lower dimensional data. These are all important avenues for additional research. To maintain focus, we address your comment in the discussion specifically with regards to your above suggestion:

Quote: “The XGBoost models performed less well in traits that had a large number of candidate SNPs selected by LASSO, likely due to overfitting. A potential approach to address this is to force the ensemble model to select less SNPs into the XGBoost model by applying LASSO over a smaller set of SNPs by imposing a stricter p-value threshold on the SNPs provided to the LASSO step, or by considering a narrower range of potential penalization parameter values for LASSO, corresponding to less selected SNPs. It is a topic of future work to assess such approaches, and to evaluate the potential trade-off, ideally in simulation studies, between the included proportion of potential SNPs used in the prediction model and prediction accuracy, while accounting for potential overfitting.”

Page, Line: page 21 lines 5-13

REVIEWERS' COMMENTS:

Reviewer #1 (Remarks to the Author):

I would like to thank the authors for addressing all of my comments, including previous ones. Great work! I have no further comments.

Reviewer #2 (Remarks to the Author):

The manuscript can now be accepted for publication from my side.

Reviewer #3 (Remarks to the Author):

In this review, the authors have provide updates to their XGBoost model. Given that the authors have reached out to Florian, and have included the relevant limitations in the limitation, I have no further major comments.

1. Page 13, under section PRSice, LDpred2 and lassosum linear PRS results, it might be useful to state that it is the relative increase in prediction performance. I guess it is generally best to mention it is a relative increase in area where it stated the performance metric, just as a reminder to the reader and to prevent misinterpretation.
2. Am quite surprise that the LDpred2 auto works better than LDpred2-inf and grid, considering that it was not testing multiple hyper-parameters. This is a relatively interesting result.
3. I wonder if the performance difference in the lassosum and LASSO selected SNPs is due to intrinsic assumption of the additive model in the part of lassosum.

Item-by-item response to final review comments

Comment: we now refer to one of the PRSs, the one using the “lassosum” method, as “lassosum2”, because it is more accurate. It was based on a different software implementation compared to the original software reported in its paper.

Reviewer 3’s comments:

1. Page 13, under section PRSice, LDpred2 and lassosum linear PRS results, it might be useful to state that it is the relative increase in prediction performance. I guess it is generally best to mention it is a relative increase in area where it stated the performance metric, just as a reminder to the reader and to prevent misinterpretation.

Response: Agreed, and we now made it clearer by adding a statement, which now reads (new text in bold):

“We compared the PVE of the best-performing PRSice-based PRS to the best-performing LDpred2 and lassosum PRS in linear PRS models. **Measured in relative PVE increase, LDpred2 performed**”.

We also reviewed other results to make sure that we use the word “relative” when reporting relative PVE change, and that it is clear when we are reporting PVE that is not relative.

2. Am quite surprise that the LDpred2 auto works better than LDpred2-inf and grid, considering that it was not testing multiple hyper-parameters. This is a relatively interesting result.

Response: The performance of LDpred2 grid may suffer from overfitting while the inf model may be somewhat mis-specified. We saw a similar pattern in another work in progress. Your comment also brought up the need to address different PRS constructions in the discussion. We added the following paragraph (page 12):

“Because we compared a non-linear ML model to a linear PRS model, we included a step where we constructed PRSs using multiple methods: clump and threshold approach implemented using PRSice, and model-based LDpred2 and lassosum2. This is an important comparison as it is not yet clear what is an optimal approach for PRS construction when using summary statistics from GWAS based on a population with different ancestral make-up compared to the target population. PRSice-based PRSs were relatively robust to the selection of clumping parameters, however, for most traits PRSice PRSs were inferior to the best PRSs from other approaches when evaluated on the held-out test dataset. In contrast, LDpred2 performance varied substantially when using its various implementations: inf, auto, and grid. LDpred2-auto had better performance than its counterparts. Possible explanations are that the grid implementation overfitted (the best performing parameter combination in the training dataset may not have been ideal for the test dataset), and that the inf model is mis-specified. Lassosum2 tended to have superior performance compared to other PRSs. We note that while lassosum2 approximates a LASSO regression, the results when using lassosum2 are different than the

results when using LASSO. This is likely due to two reasons, First, the selection of SNPs used: because standard LASSO implementation cannot handle, computationally, too many SNPs, we implemented it using SNPs with $p\text{-value} < 10^{-4}$ and further divided into five sets of SNPs. In contrast, lassosum was implemented using 1M SNPs with the lowest p-values in the summary statistics (and that were also available in our dataset), without clumping. Second, the lassosum model assumes that the marginal SNP effect sizes are as supplied by the GWAS summary statistics, while the LASSO model does not have such an assumption and it only relies on the available individual-level data.”

3. I wonder if the performance difference in the lassosum and LASSO selected SNPs is due to intrinsic assumption of the additive model in the part of lassosum.

Response: this is likely because (a) we started from a different set of SNPs: Because LASSO is computationally limited, we only considered SNPs with $p\text{-value} < 10^{-4}$, but the candidate SNPs in lassosum were different. (b) the lassosum model assumes that the marginal SNP effect sizes are as supplied by the GWAS summary statistics, while the LASSO model does not have such an assumption and it only relies on the available individual-level data. We now explain this in the discussion in the paragraph above (see response to comment 2).